



# Northern Hemisphere atmospheric history of carbon monoxide since preindustrial times reconstructed from multiple Greenland ice cores

Xavier Faïn[1], Rachael H. Rhodes[2], Philip Place[3], Vasilii V. Petrenko[3], Kévin Fourteau[4],
Nathan Chellman[5], Edward Crosier[3], Joseph R. McConnell[5], Edward J. Brook[6], Thomas Blunier[7],
Michel Legrand[8], and Jérôme Chappellaz[1]

[1]Univ. Grenoble Alpes, CNRS, IRD, Grenoble INP, IGE, 38000 Grenoble, France
[2]Department of Earth Sciences, University of Cambridge, Cambridge, CB2 3EQ, UK
[3]Department of Earth and Environmental Sciences, University of Rochester, Rochester, NY, USA
[4]Univ. Grenoble Alpes, Université de Toulouse, Météo-France, CNRS, CNRM, Centre d'Études de la Neige, Grenoble,
France
[5]Division of Hydrologic Sciences, Desert Research Institute, Reno, NV 89512, USA
[6]College of Earth, Ocean, and Atmospheric Sciences, Oregon State University, Corvallis, OR 97331, USA
[7]Centre for Ice and Climate, Niels Bohr Institute, University of Copenhagen, Copenhagen, Denmark
[8]LISA, UMR CNRS 7583, Université Paris-Est Créteil, Université de Paris, Institut Pierre Simon Laplace (IPSL), Créteil,
France

**Abstract.** Carbon monoxide (CO) is a regulated pollutant and one of the key components determining the oxidizing capacity of the atmosphere. Obtaining a reliable record of atmospheric CO mixing ratios since pre-industrial times is necessary to evaluate climate-chemistry models in conditions different from today and to constrain past CO sources. We present high-resolution measurements of CO mixing ratios from ice cores drilled at five different sites on the Greenland ice sheet which experience

a range of snow accumulation rates, mean surface temperatures, and different chemical compositions. An optical-feedback cavity-enhanced absorption spectrometer (OF-CEAS) was coupled to continuous melter systems and operated during four analytical campaigns conducted between 2013 and 2019. Overall, continuous flow analyses (CFA) of CO were carried out on over 700 m of ice. The CFA-based CO measurements exhibit excellent external precision (ranging 3.3-6.6 ppbv, $1\sigma$), and achieve consistently low blanks (ranging from 4.1±1.2 to 12.6±4.4 ppbv), enabling paleo-atmospheric interpretations. However the

five CO records all exhibit variability too large and rapid to reflect past atmospheric mixing ratio changes. Complementary tests conducted on discrete ice samples demonstrate that these variations are not artifacts of the analytical method (i.e., production of CO from organics in the ice during melting), but very likely are related to in situ CO production within the ice before analysis. Evaluation of signal resolution and co-investigation of high-resolution records of CO and TOC show that past atmospheric CO variations can be extracted from the records' baselines at four sites with accumulation rates higher than 20 cm water equivalent

per year (weq yr$^{-1}$). However, such baselines should be taken as upper bounds of past atmospheric CO burden. Baseline CO records from four sites are combined to produce a multisite average ice core reconstruction of past atmospheric CO for the Northern Hemisphere high latitudes, covering the period from 1700 to 1957 CE. From 1700 to 1875 CE, the record reveals stable or slightly increasing values in the 100-115 ppbv range. From 1875 to 1957 CE, the record indicates a monotonic increase from 114±4 ppbv to 147±6 ppbv. The ice-core multisite CO record exhibits an excellent overlap with the atmospheric





CO record from Greenland firn air which spans the 1950-2010 time period. The combined ice-core and firn air CO history, spanning 1700-2010 CE provides useful constraints for future model studies of atmospheric changes since the preindustrial period.

## 1    Introduction

Carbon monoxide (CO) is a reactive trace gas that plays a crucial role in the interactions between climate and atmospheric
chemistry. CO strongly affects the global oxidative capacity of the atmosphere by acting on the budgets of both hydroxyl radical (OH) and ozone ($O_3$). CO is the principal sink for the tropospheric OH, with up to $40\%$ of the OH radicals reacting with CO in the modern troposphere (Lelieveld et al., 2016). Thus, CO indirectly impacts the lifetime of numerous atmospheric constituents such as methane (CH4), non-methane hydrocarbons (NMHCs), and hydrofluorocarbons (HCFCs). Oxidation of CO by OH ultimately leads to CO2 production, and in the presence of high levels of nitrogen oxides (NOx) can also result in
significant production of tropospheric ozone (Crutzen, 1973). Since CO impacts the atmospheric budgets of greenhouse gases such as $CO_2$, $CH_4$ and tropospheric ozone, its present-day emissions lead to an indirect radiative forcing of about 0.23±0.05 W $m^{-2}$ (Myrhe et al. 2013). Furthermore, CO has another indirect climatic impact through the formation of submicron secondary organic aerosol produced by oxidation of volatile organic compounds (VOCs).

CO is emitted by various surface processes and produced by atmospheric oxidation of different gaseous precursors. Atmo-
spheric oxidation of $CH_4$ and VOCs represents about half of the sources (Duncan et al., 2007). The surface sources include incomplete combustion of anthropogenic fossil fuels and biofuels (Hoesly et al., 2018), biomass burning (van der Werf et al., 2017), and minor contributions from plant leaves (Tarr et al., 1995; Bruhn et al., 2013) and the ocean (Conte et al., 2019). The oxidation by OH is the dominant sink of CO, which results in a mean global CO tropospheric lifetime of about 2 months (Khalil et al., 1999). However, the CO lifetime strongly varies with latitude and season, ranging from 20–40 days in the tropics
and up to 3 months in polar areas (Duncan et al., 2007).

The understanding of the past CO trends is crucial for monitoring how anthropogenic activities have impacted CO emissions, atmospheric chemistry and composition. Over the last decades, analyses of past CO trends have been made possible by a growing number of direct and indirect observations. Monitoring of atmospheric CO, initiated by NOAA (National Oceanic and Atmospheric Administration) in 1988, revealed a global decrease of tropospheric CO by ∼0.5 ppbv $yr^{-1}$ between 1988 and
1996 (Novelli et al., 1998). The space-born MOPITT (Measurements of Pollution in The Troposphere instrument) has monitored the atmospheric CO burden since 2000 (Deeter et al., 2017), and reveals a declining trend. This trend is more pronounced in the Northern Hemisphere (Zheng et al., 2019) which concentrates more sources related to the global economic activity such as fossil fuel use and CO emission from combustion processes.

Ground based and satellite-derived CO data are only available, however, for the last three decades. Ancient air preserved in
glacial ice and firn is thus a unique archive for reconstructing the past atmospheric CO record prior to the 1990s. The firn is the upper layer of an ice sheet where snow is slowly transformed into ice. A large amount of air can be sampled from the interconnected open pores. Mean ages of atmospheric gas increase with firn depth. CO has only been measured in firn air at a



few Northern Hemisphere sites, including Summit, the North Greenland Ice Core Project (NGRIP), and the North Greenland
Eemian drilling site (NEEM) (Petrenko et al., 2013). Measurements performed on the Devon Island (Canadian Arctic) firn
air showed CO artefacts precluding atmospheric reconstruction (Clark et al., 2007). The three central Greenland firn records
allowed for a 60 yrs reconstruction of atmospheric CO and revealed that atmospheric CO mixing ratios in the Arctic following
industrialization increased from 145 ppbv in 1950 CE (Common Era) until around the late 1970s with a peak of 155 ppbv, and
decreased to 140 ppbv in the late 1990s. This decreasing trend in atmospheric CO mixing ratio reconstructed from firn archive
is in agreement with direct observations (ground and satellite based) for the last decades. Isotopic measurements of the NEEM
firn air suggested this pattern was driven mainly by a change in CO emissions derived from fossil fuel combustion (Wang et al.,
2012).

Analysis of air trapped in bubbles in solid ice below the firn layer is required to extend such reconstruction further back in time,
beyond 1950 CE. The pioneering measurements conducted on the Eurocore ice archive (Haan et al., 1996; Haan and Raynaud,
1998) found that Northern Hemisphere (NH) [CO] increased gradually from ∼90 ppbv in 1850 CE to 110 ppbv in ∼1950
CE and that CO levels were stable at ∼90 ppbv from 1625 to 1850 CE. Air older than 1600 CE (below 167 m depth) exhib-
ited higher variability and elevated CO levels (100–180 ppbv) with parallel anomalies in the $CO_2$ record. This section of the
Greenland Eurocore record is thought to reflect in situ CO production rather than an atmospheric signal. Haan and colleagues
determined CO concentrations along the Eurocore archive using gas chromatography, including a mercuric oxide detector,
combined with wet extractions of the trapped gases from discrete pieces of ice. This protocol was affected by a relatively large
blank (typically 10 ppbv), was time consuming and resulted in limited resolution reconstructions.

Over the last decade, Continuous Flow Analyses (CFA) of ice core $CH-4$ concentrations utilising laser spectroscopy (Stowasser
et al., 2012) has become a widely-used tool in palaeoclimatology (e.g., Rhodes et al., 2015). CFA was applied for the first time
to CO with the measurement from the NEEM-2011-S1 core (Faïn et al., 2014), as an attempt to reconstruct an atmospheric
history for NH CO over the past 1800 yrs. The ultra-high resolution of CFA analysis means it is a powerful tool to distinguish
between depth sections impacted by in situ CO production and sections preserving the atmospheric record. Faïn et al. (2014)
report stable measurements of CO mixing ratios with an external precision of 7.8 ppbv ($1\sigma$), but a poorly constrained procedu-
ral blank and poor accuracy because absolute calibration was not yet possible. The NEEM-2011-S1 CO mixing ratios reported
are highly variable throughout the ice core with high frequency (at the annual scale), high amplitude spikes characterizing
the record. These CO signals likely result from in situ production occurring within the ice itself, with patterns too abrupt and
rapid to reflect atmospheric variability. In situ production coupled with the procedural blank and accuracy problems largely
prevented interpretation of the record in terms of atmospheric CO variations.

In this study, we expand on this exploratory investigation by reporting continuous CO data measured on a set of five additional
Greenland ice cores. By combining the analysis of these new ice core records, and separating the high frequency CO signals
driven by in situ production from baseline concentrations, we reveal in the Arctic and for the last three centuries (i) an up-
per bound estimate of past atmospheric CO burden and (ii) atmospheric [CO] trends. Climate-chemistry models and/or Earth
System Models can produce simulated atmospheric [CO] at ground level in Central Greenland, from the preindustrial era to
present-day. Such models are presently intercompared within the AerChemMIP exercise (Collins et al., 2017). The compar-





ison of the past evolution of Arctic atmospheric CO mixing ratio extracted from Greenland ice archives in the frame of this study with AerChemMIP model outputs is out of the scope of this paper. Such comparison, which should also allow to better
constrain CO emissions inventories of emissions factors, will be addressed in a future study.

## 2    Methods

### 2.1    Sample description

Five ice cores extracted from Greenland (Table 1) were investigated in this study (Fig. S1). High resolution CO mixing ratio data were measured continuously along with those of methane. The methane data for the NEEM-SC, D4, North Greenland
Ice Core Project (NGRIP), and Tunu13 sites have been reported previously (Rhodes et al., 2016). The NEEM-SC section was chosen to extend the existing NEEM-2011-S1 record (Faïn et al., 2014; Rhodes et al., 2013) further back in time, and a composite NEEM signal including the NEEM-2011-S1 and new NEEM-SC sections is presented here. The calibration of the NEEM-2011-S1 CO data was revisited before building this composite (see Supplementary Information, SI).

The fifth ice core in this study is a new core, retrieved from central Greenland during June 2015: the PLACE ice core. The
PLACE core was drilled 1 km away from the location of Eurocore site(72.58°N and 37.64°W, drilled 1989; Schwander and Rufli, 1994), which in prior work was suggested to contain an intact atmospheric CO signal for air ages younger than 1650 CE (Haan et al., 1996; Haan and Raynaud, 1998). Care was taken when selecting the PLACE site to avoid drilling in areas that were disturbed during the Eurocore and later GRIP ice core drilling operations. The core sections were drilled with the Blue Ice Drill (BID) (Kuhl et al., 2014) and immediately placed in the shade in a clean snow area after drilling, before packing in ice
core boxes. These boxes were then stored in a snow cave (at -20°C or colder). At the end of the season, the cores were removed from the snow cave at the field site and returned to Summit Station for transport to the National Ice Core Laboratory (NICL), Denver, CO USA. Core temperature was verified to not have exceeded -15°C during transport by including temperature loggers in some of the ice core boxes.

The five sites investigated experience accumulation rates ranging from 8 to 41 cm weq yr$^{-1}$, mean annual surface temperature
between -24 and -32°C, and different chemical composition (Table 1, Sect. 3.3). Description of the ice and gas chronologies for the five ice cores is reported in Table S1.

### 2.2    CO continuous flow analyses

Since the first continuous, high resolution, CO measurements were performed along the NEEM-S1 ice core in 2011 (Faïn et al., 2014), CFA-based CO analyses have greatly improved, including lowering the CO blank and characterizing how CO is pref-
erentially dissolved during the CFA process so as to establish an absolute calibration. The excellent precision of CO analyses has been confirmed, and the designs of CFA setups themselves have been optimized to limit the instrumental smoothing and improve signal resolution. This section reviews the operation and recent improvements of CFA-based CO measurements. More details can also be found in Supplementary Information (SI).



**Table 1.** Locations, site characteristics and other relevant information for ice cores featured in this study.

| Ice core and location | Depth interval (m) | Gas age interval (yrs CE) | Accum. Rate (cm ice yr$^{-1}$) | Mean annual Temp. (°C) |
|---|---|---|---|---|
| D4 <br> Central Greenland <br> 71.40°N, 43.08°W <br> 2713 m elevation | 63-145 | 1825;1961 | 41[a] | -24[a] |
| Tunu13 <br> NE Greenland <br> 78.03°N, 33.88°W <br> 2200 m elevation | 61-213 | 836 ; 1893 | 10-14[a] | -29[b] |
| NGRIP <br> Central Greenland <br> 75.10°N, 42.32°W <br> 2917 m elevation | 74-108 <br> 207-254 <br> 519-569 <br> 1215-1265 | 1780 ; 1926 <br> 980 ; 1237 <br> -929 ; -616 <br> -5545 ; -5899 | 19[c] | -31.5[c] |
| PLACE <br> Central Greenland <br> 72.58°N, 37.64°W <br> 3200 m elevation | 80-153 | 1447; 1957 | 20.5 | -32 |
| NEEM-2011-S1 <br> NW Greenland <br> 77.45°N, 51.06°W <br> 2450 m elevation | 71-409 | 270 ; 1961 | 22[d] | -28.9[d] |
| NEEM-2011-SC <br> NW Greenland <br> 77.45°N, 51.06°W <br> 2450 m elevation | 399-573 | -682 ; 322 | 22[d] | -28.9[d] |

[a]Rhodes et al. (2016); [b]Butler et al. (1999); [c]community members (2004); [d]NEEM community members (2012).

### 2.2.1 System operation

The ice cores listed in Table 1 were analysed using continuous ice core melter systems coupled with online gas measurements. The general principles of this analytical approach are provided by (Stowasser et al., 2012). Briefly, ice core sticks are cut at a 34 mm x 34 mm cross section and processed on a melterhead located in a cold room. The melterhead is composed of inner and outer collection areas with the inner area dedicated to sample collection. To prevent contamination, an overflow from the inner



to the outer melterhead areas of >10% is created by lowering the sample pumping speed. The water and gas bubble mixture
is continuously pumped via a debubbler into a temperature-controlled gas extraction unit maintained at 30°C. The gas/water
volume ratio of the sample is about 10% before the debubbler, and 50% after the debubbler. A fully degassed fraction of the
melted sample is thus also available at the debubbler for complementary chemical analyses in the liquid phase. The gas is
extracted along the sample line after the debubbler by applying a pressure gradient across a gas-permeable membrane, either
Transfer-Line (Idex) or Micromodule 0.5×1 (Membrana GmbH) degassers. Then, the gas is dried by a custom-made Nafion
(Perma Pure) dryer. Finally, CO (and/or $CH_4$) mixing ratios are continuously monitored along the gas sample flow by a laser
spectrometer.

Our samples were all analysed at the Desert Research Institute, Reno (NV, USA) during two analytical campaigns (Tunu13,
D4, NEEM-SC, and NGRIP in 2013; PLACE in 2015). The NEEM-2011-S1 core was analysed at DRI in 2011, as described
by Faïn et al. (2014). The PLACE core was also analysed at IGE (Grenoble, France) in 2017. Specific descriptions of the DRI
and IGE setups were already provided by Rhodes et al. (2015) and Fourteau et al. (2017), respectively, and a comparison of
the two setups is reported in the SI.

A unique spectrometer, using optical feedback cavity enhanced absorption spectrometry (SARA, developed at Laboratoire
Interdisciplinaire de Physique, University Grenoble Alpes, France; Morville et al., 2005) was used to analyse carbon monoxide
(and simultaneously, methane) at both DRI and IGE and during the three analytical campaigns. Detailed description of this
instrument, which was used for CO measurements along the NEEM-2011-S1 core in 2011 at DRI, is reported by Faïn et al.
(2014). The OF-CEAS instrument was always carefully calibrated against standard gases (Table S2) before melting ice cores
(see SI).

### 2.2.2 Calibration Loop

The degassing membranes operated so far in gas-CFA systems extract bubbles from the sample flow but do not recover dis-
solved gases from the water phase efficiently. Carbon monoxide (or methane) have higher solubility than $N_2$ or $O_2$. Conse-
quently, mixing ratios of CO in the gas phase of the sample flow exhibit lower values than initially existing in the ice bubbles.
These offsets need to be corrected for, and an attempt to replicate the conditions experienced by the ice core water-gas mixture
between the melterhead and the laser spectrometer was reported originally by Stowasser et al. (2012). As melting ice contains
10% air by volume, a 10:90 mixture of synthetic air and degassed deionized (DI) water can be introduced into the system via
a 4-port valve located directly after the melterhead. The water is sourced from a 2 L reservoir degassed by constantly bubbling
ultra-pure He through it. The air–water mixture follows the same path through the system as the ice core sample before being
analysed by the laser spectrometer. However, it is not completely identical as it includes more components such as an additional
peristaltic pump or extra lines. This pathway for synthetic standard analysis will be referred to as "calibration loop" here, but
has been designated as "full loop" in previous studies (e.g., Rhodes et al., 2013; Faïn et al., 2014). Note that this "calibration
loop" should not be confused with the "Internal Loop" (Rhodes et al., 2013) a system to isolate the gas extraction system from
the remainder of the melting system, while keeping a continuous gas extraction (see Fig. 1 from Rhodes et al., 2013).



### 2.2.3 CO blank of CFA-based analyses

The CO procedural blank was evaluated for each analytical campaign. Two different evaluation approaches were applied, which are described in SI. A relatively high CO blank of 35±7 ppbv was observed at DRI in 2013, when using a Membrana

Micromodule degasser for analysing the Tunu13, D4, and a fraction of the NEEM-SC cores. Lower blank values, ranging 4.1±1.2 to 12.6±4.4 ppbv were observed when using an Idex Transfer-Line degasser (cores PLACE, NGRIP, a fraction of NEEM-SC).

### 2.2.4 Internal precision and stability

Internal precision and stability of gas-CFA measurements can be evaluated by Allan variance tests (Allan, 1966) applied to the

calibration loop dataset. Laser spectrometers produce many measurements per minute, with a data acquisition rate of 6 Hz for our OF-CEAS instrument. Observed optimal integration time (i.e., time of lowest Allan deviation) was determined for each analytical campaign, and is now larger than 500 s for DRI, and 1000 s for IGE CFA setups (SI), which is much longer than the 5 s reported by Faïn et al. (2014) and testament to increased stability. However, to maximize the depth resolution, CO data can be averaged over shorter integration time (IT). There is a trade-off between internal precision and IT. Abrupt, non-climatic

signals in gas records, such as in situ CO production (Faïn et al., 2014), cannot be fully resolved without reducing the IT. Internal precision, defined as twice the Allan deviation at chosen integration times, ranged from 1.2 to 1.6 ppbv depending on the analytical campaign (SI and Table S2).

### 2.2.5 External precision

External precision of the continuous CO measurements (i.e., including all sources of errors or bias) was investigated by melting

replicate ice sticks on different days during each analytical campaign. Specifically, we defined the external precision as the pooled standard deviation calculated on the differences of CO concentrations from main and replicate analysed ice sticks, averaging continuous CO data over few cm long intervals. This approach estimates an external precision of 5.7 ppbv for the DRI gas-CFA setup in 2011 (NEEM-2011-S1 campaign, 18 m long replicated section, using 9 cm intervals). We reproduced this analysis for the 2015 DRI campaign, and established an external precision of 6.6 ppbv (19.6 m long replicated section,

10-cm long intervals, Fig. S3). With a similar methodology, but a shorter replicate section analysed, we evaluated the external precision of PLACE continuous CO measurements conducted at IGE in 2017 to 3.3 ppbv (2.3 m long replicated section, 1-cm intervals, Fig. S3). No replicate ice core sections were available and measured during the 2013 DRI campaign. For the 2013 campaign, we consider the external precision to be 5.7 ppbv for the ice cores analyzed in 2013 with a Micromodule and 6.6 ppbv using the Idex Transfer-Line degasser, by extrapolating results from similar CFA setups.

### 2.2.6 Absolute calibration and accuracy

CFA–based gas records must be corrected for under-recovery of gases dissolved in the water stream to obtain absolute values on the WMO-CO X2014A calibration scale. The magnitude of gas dissolution is monitored using the calibration loop (Sect.



2.2.2). In this study, a 73 m long ice core (PLACE) was analysed on two different CFA setups (at IGE and DRI), for CO but also for $CH_4$ mixing ratios. This gas CFA laboratory intercomparison revealed that the fraction of $CH_4$ not recovered at IGE was larger than at DRI (14% and 10%, respectively). Note that the gas extraction unit and OF-CEAS detection instrument were identical, and only the melter and lines upstream of the Idex Transfer-line degasser had different geometries (Sect. 2.2.3). This intercomparison demonstrates that the solubility-based calibrations are required to transfer gas CFA datasets into absolute concentration scales, and are dependent on the geometry and operation of a CFA setup. This observation was further confirmed at IGE by measuring the $CH_4$ mixing ratio at IGE when melting replicate ice sticks with varying CFA system geometries (eg., length of lines) and operation of the CFA setup (e.g., melting speed, pumping rate). For each analytical campaign, the geometry and operational protocol of the CFA setup were thus kept unchanged during the measurements. However, these observations rule out universal modelling of the preferential CO dissolution by using Henry's Law coefficient, and demonstrate that CFA setups are dynamic systems, not at solubility equilibrium.

In this study, we used the calibration loop for absolute calibration of the CO dataset. Such an approach has been successfully applied for $CH_4$ (e.g., Rhodes et al., 2015). However, calibration loops need to be carefully set up and operated, to closely reproduce solubility losses occurring in the CFA sample pathway. In the case of $CH_4$, discrete datasets often allow for an external validation of this CFA internal calibration, but such validation was not available for Greenland CO. We hypothesise that CO and $CH_4$ dissolution follow the same physical laws: consequently, if a calibration loop is able to reproduce methane preferential dissolution, it should also reproduce CO losses related to dissolution. Thus, we extracted from calibration loop experiments for each analytical campaign, a solubility calibration (SC) factor for CO. Overall, CO losses driven by preferential dissolution ranged from 4 to 9%. Replicate measurements show that the fraction of CO not recovered at the outlet of the CFA system was very stable during analytical campaigns, both at DRI and IGE. Based on repeated calibration loop measurements throughout the campaigns we conservatively estimate the uncertainty of the SC factor to be ±1% (2 sigma).

To further evaluate the robustness of the CFA procedure and this absolute calibration approach, five discrete samples of the PLACE core were analysed with a discrete CO setup (SI Sect. 1.7.4, Fig. S7). The agreement between discrete and CFA-based CO dataset (both IGE and DRI) was excellent, with differences in CO mixing ratio ranging from 0.4 to 2.8%. The depth intervals of the discrete samples encompassed the entire span of the PLACE core, from 84 to 146 m depth. More details about absolute CO calibration are reported in Supplementary Information.

### 2.2.7 Signal Smoothing

The mixing of gases and meltwater during the sample transfer from the melt head to the laser spectrometer induces a CFA experimental smoothing of the signal. The extent of the CFA-based damping was determined for each analytical campaign by performing step tests, i.e., switches between two synthetic mixtures of degassed DI water and synthetic air standards of different $CH_4$ concentrations (Stowasser et al., 2012, see SI). A cut-off wavelength can be defined as the wavelength of a sine signal experiencing a 50% attenuation in amplitude. Cut-off wavelengths from 15.0 cm during the DRI 2013 campaign (when using Membrana Micromodule degasser) to 1.6 cm during IGE 2017 campaign (Table S2). Lower cut-off wavelengths were



obtained by limiting system dead volume, most notably by reducing the debubbler internal volume and by using an optimized Idex Transfer-Line degasser instead of Membrana Micromodule degasser (1 and 5 ml internal volume, respectively)

### 2.2.8 Data processing

CO and ice chemistry data collected at DRI were mapped onto depth scales using high-resolution (0.1–0.5 Hz acquisition rate)
liquid conductivity data and time–depth relationships recorded by system operators. A constant melt rate for each metre length of core is assumed. DRI depth scale uncertainties are estimated to be $\pm 2$ cm ($2\sigma$).

Occasional entry of ambient air into the analytical system as breaks in the core were encountered caused contamination. The OF-CEAS spectrometer simultaneously measures CO and $CH_4$ mixing ratios, and such contamination was characterised by a sharp increase in $CH_4$ concentration (ca. 1900 ppbv) followed by an exponential decrease. Data were manually screened for
these ambient air contamination events.

Uncertainty on CO mixing ratios is established as $1\sigma$ and calculated for each calibrated CO dataset. It combined uncertainties evaluated specifically for each analytical setup on CO blanks, solubility calibration factors, and internal precision of CO CFA measurements (see SI and previous sections).

Finally, we used high resolution water isotope datasets to match DRI and IGE PLACE depth scales. Water isotopic ratios were
measured simultaneously with gas concentrations in both laboratories using laser spectroscopy (Gkinis et al., 2011; Maselli et al., 2013).

### 2.3 Investigation of possible rapid CO production from trace organics in the ice during melting

To investigate if elevated or highly variable carbon monoxide levels observed in Greenland ice core could originate from the melting process (i.e., "in extractu" CO production from trace organics in the ice), CO concentrations were determined in
tests with 20 discrete ice and firn samples (sized between 272 - 1089 g, average size of 690 g). The experimental protocol is reported in detail in the SI. Briefly, ice samples collected below close-off depth (fully closed porosity) are completely grated to fine powder to remove trapped air. Firn samples do not contain a significant amount of trapped air; however some firn samples were still grated to powder to verify that the grating process did not increase the [CO] blank. The samples are then introduced in a glass vessel, evacuated and flushed with ultrapure air overnight. Gas standard of known concentration is then introduced
within the vessel. Consequently, for both firn and ice samples CO mixing ratio within the vessel prior to melting is equal to that of the gas standard introduced in the vessel. If CO in extractu production occurs during melting, it should cause an increase in CO mixing ratio above this known value.

The CO concentration in the gas available in the vessel headspace is determined by gas chromatography combined with a mercuric oxide reduction detector (RGD; Peak Performer 1 from Peak Laboratories), at three different stages: before melting,
during melting, and after melting. CO concentrations are calibrated using four synthetic air gas standards with nominal CO concentrations ranging from 50 to 500 ppbv, and data are reported on the WMO-CO X2014A scale.

Three large diameter (27 cm) cores collected with the Blue Ice Drill (BID) were used for these tests: the PLACE ice core (Central Greenland, Table 1), the C14 firn core (Central Greenland, drilled in June 2015, 72°39.62' N, 38°34.85' W), and a



Taylor Glacier ice core (Antarctica, drilled in December 2015, 77°45.69' S, 161°43.18' E). BID cores allowed for up to four

true depth replicates to be measured from the same depth interval. CO concentrations were determined (before, during, and after melting) for the following cases: (i) Greenland C14 firn ( 45 m depth), (ii) Greenland PLACE ice from depths ranging 110-125 m and showing low and stable [CO] on the continuous CFA record, (iii) Greenland PLACE ice from depths ( 111 m) showing elevated and highly variable [CO] on the continuous CFA, (iv) Oldest Dryas ( 15 kyrs BP) and Holocene ( 9.5 kyrs) Antarctica ice, and (v) gas-free ice made from degassed Milli-Q 18.2MΩ.

**2.4 Chemistry data**

During the two DRI analytical campaigns, melted ice core samples were analysed continuously by inductively coupled plasma mass spectrometry (ICP-MS) and CFA for chemical species. These analytical methods have been reported previously (McConnell and Edwards, 2008; McConnell et al., 2007, 2002). High-resolution measurements of total organic carbon (TOC) were obtained at DRI by coupling a Sievers 900 TOC analyzer to the DRI ice core melter (Legrand et al., 2016).

**3 Results and Discussion**

The new CO records available from the PLACE, NGRIP, NEEM, D4, and Tunu13 ice-cores are reported in Fig. 1 and plotted for the period spanning 1650 to 1960 CE (gas age, see Table S1), the full records being reported in Fig. S15. Over the 1650-1960 CE period, all Greenland CO ice records exhibit highly variable CO values. A minimum mixing ratio of 92 ppbv is observed at NEEM and PLACE, and a maximum mixing ratio of 1191 ppbv at Tunu13. Considering all records, some 46 events revealed

mixing ratios higher than 250 ppbv. In Sect. 3.1 we discuss the baseline CO levels of the different ice-core records, and the characteristics of abrupt CO spikes detected in all records. Tests and discussions presented in Sect 3.2. demonstrate that in situ production drives these abrupt CO spikes. The key question of whether an atmospheric signal can be extracted from the low frequency variability of the CO ice-records' baselines is discussed in Sect. 3.3. Comparison of our new CO records with existing data is presented in Sect. 3.4. Finally, a multi-site reconstruction of past atmospheric CO levels at northern latitudes is

presented in Sect. 3.5.

**3.1 Continuous CO records along Greenlandic ice cores**

Figure 1 shows that the five Greenlandic sites investigated in this study all exhibit highly variable CO mixing ratios. Referring to previous data reported by Faïn et al. (2014), such a pattern was expected when measuring the NEEM-SC samples and extending the NEEM-2011-S1 record. However, new records from central Greenland (PLACE, NGRIP), north-eastern Greenland

(Tunu13) and southern Greenland (D4) appear to be also affected by variability in CO mixing ratios that is too large and rapid to reflect past atmospheric mixing ratio changes (Figs. 1 and S15).

The true extent of high CO variability may be masked, however, by CFA analytical smoothing. Figure 2 (upper panel) reports a 1.5 m long section of the PLACE ice-core, analysed with both DRI and IGE CFA setups. The higher CFA setup resolution at IGE (Sect. 2.2.7) reveals more features, but may still miss even higher frequency variability and dampen extrema values.



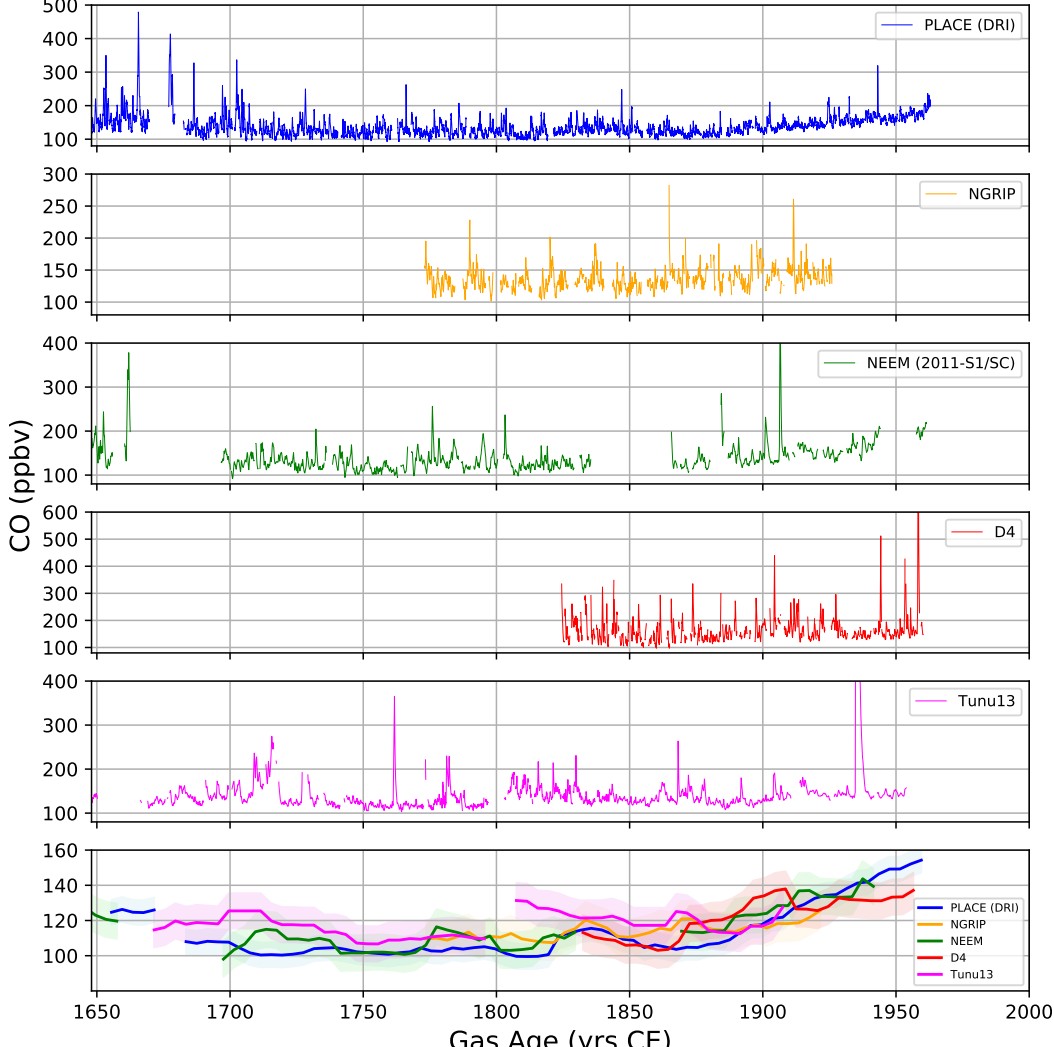

**Figure 1.** Upper panels: Greenland continous CO records available from the PLACE (DRI), NGRIP, NEEM, D4, and Tunu13 ice-cores and plotted for the period spanning 1650-1960 CE. Lower panel : CO baseline levels from each ice-record defined as the $5^{th}$ percentile of data every 4 years over a moving window of 15 yrs, with shaded envelopes reported on each baseline representing the uncertainty.





Similarly, a CO record from a core drilled at a relatively low snow accumulation site will appear more smoothed out by CFA
analysis than one from a relatively high snow accumulation rate site. Therefore, in this study we do not directly compare the
amplitude of CO variability between D4 and Tunu13, which exhibit a factor of four difference in snow accumulation rate.

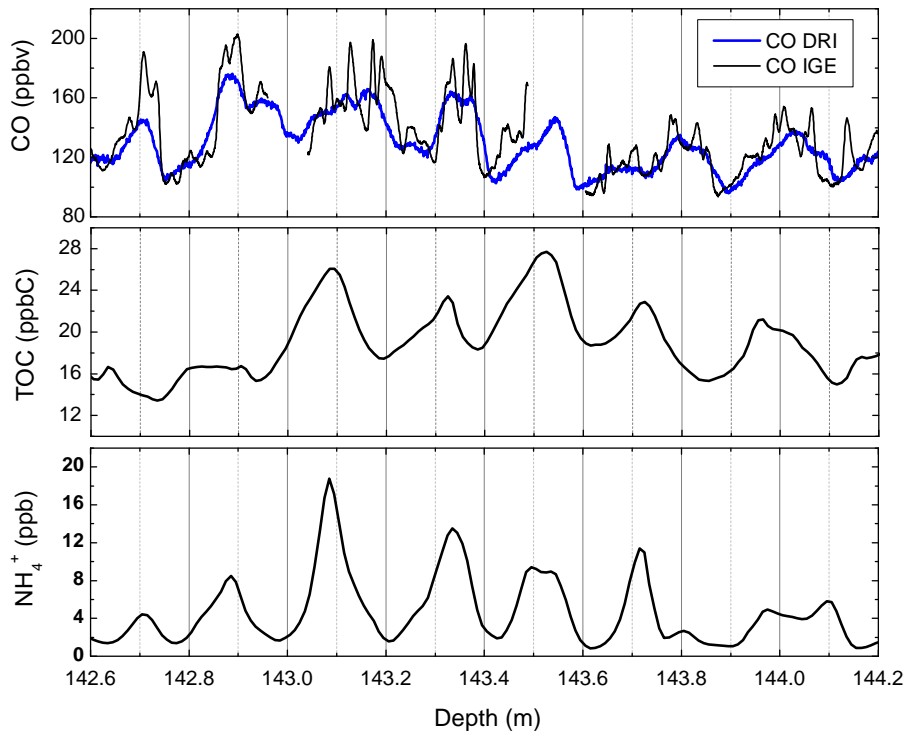

**Figure 2.** High-resolution TOC, $NH_4^+$, CO from the PLACE core. CO mixing ratios were measured with both the DRI and IGE CFA systems.
$NH_4^+$ shows a well-marked seasonal cycle characterized by a summer maximum. PLACE snow accumulation is about $20\,cm\,weq\,yr^{-1}$
.

High frequency variability within CO records can be quantitatively characterized using MAD (Median Average Deviation).
MAD was calculated for all CO records collected at DRI every 4 yrs over a moving window of 15 yrs (similarly to baseline
signals, see Sect. 3.1.1). We observe for the 1700-1950 CE time period that all records exhibit stable and similar MAD values.
Integrated MAD values calculated for the entire 1700-1950 CE period range 9.9-14.2 ppbv. The NEEM, NGRIP and Tunu13
datasets significantly extend beyond the last three centuries and reveal different MAD patterns for age older than 1700 CE.
The Tunu13 record, spanning the last 1300 yrs, reveals that CO variability remains stable as depth increases while NEEM
and NGRIP show increasing MAD with depth, clearly indicating that high variability in CO is more pronounced for older
ice. These increases are significant, with MAD values reaching $\sim 60$ ppbv for the deeper sections of the analyzed NEEM and
NGRIP ice-core sections.





### 3.2 Do drilling or analytical processes cause CO production?

The observation of high frequency CO variability along the NEEM-2011-S1 core led Faïn et al. (2014) to discuss if such patterns could be driven by the processes involved in sample collection and analysis. Here we consider our new data to assess in more detail if such processes can drive CO production.

#### 3.2.1 Ice core drilling

We observe that abrupt CO spikes of similar magnitude occur in ice-cores extracted by dry drillings (e.g., PLACE, D4, Tunu13) and when a drilling fluid was used (e.g., NEEM, NGRIP) (Fig. 1). This clearly rules out contamination from the drilling fluid as a cause for high frequency CO variability, confirming the previous conclusion from Faïn et al. (2014).

Haan et al. (2001) observed CO production within an alpine snowpack in daylight. Drillings are usually conducted at high-altitude sites in summer where UV radiation levels are high. One could thus hypothesise that CO is photochemically produced within ice-cores just after drilling, when ice-cores are handled at the drilling site before being packed in ice-core boxes. During the PLACE and NEEM-SC drillings, specific care was taken to never expose directly freshly drilled cores to sunlight. For PLACE, a shaded area was set up for core processing. NEEM 2011-S1 and SC core drillings were conducted in a tent and a trench, respectively. In spite of that, all these ice-cores reveal CO spikes. In contrast, no specific care was taken during the historical Eurocore drilling when handling cores, with respect to sunlight exposure, and Haan and Raynaud (1998) report that this archive does not exhibit highly variable CO concentration in the upper section. Based on this, it appears that brief direct exposure to sunlight after drilling is unlikely to be the cause of the large CO spikes

#### 3.2.2 Impact of long-term core storage

By comparing the storage time of different ice-cores (storage without light exposure) prior to CO analysis with the same CFA setup (NEEM-2011-S1 and D4), Faïn et al. (2014) concluded that CO production in ice during core storage was unlikely. We extend this observation by comparing central Greenland archives: NGRIP and PLACE records reveal similar abrupt CO spikes and CO baselines (Fig. 1) while PLACE and NGRIP cores were analysed 3 months and 15 years, respectively, after drilling. Furthermore, we measured at IGE a 10 m long replicate section of the PLACE core nearly two years after the main analytical campaign (respectively December 2018 and February 2017) and no change in the CO values was observed (Fig. S12).

#### 3.2.3 Impact of CFA process

In 2013 at DRI, a 4 m long replicate section of the D4 ice-core was melted with all laboratories kept in darkness. Similar in situ CO production patterns were observed under light and dark melting conditions. These results, already reported by Faïn et al. (2014, Fig. S2), reveal that CO variability observed along the D4 core are not related to photochemically driven CO production occurring in the sampling lines between the melter head and the CFA extraction box. To further evaluate if the CFA analytical system could somehow induce CO production within the melted ice, a melted D4 sample was collected downstream of the




degassing membrane and re-circulated in the calibration loop mode (Sect. 2.2.2) for one hour as degassed blank water. We observed similar CO levels to the deionised water blank.

### 3.2.4 Results of investigation of possible rapid CO production from trace organics in the ice during melting

As described above, CO analyses of 20 discrete ice and firn core samples (including the PLACE core) were carried out at the University of Rochester, with the goals of investigating the possibility of in extractu production of CO from trace organics in the ice during melting of ice core samples (Sect. 2.3 and SI, Table S4).

All of the different ice and firn sample types, including samples of different mass and a gas-free ice sample, showed a consistent excess CO growth occurring during the melting step, with excess CO increasing by 6.4 ppbv on average (Fig. 3) - a much

smaller CO enhancement than the spikes observed in the CFA records. Since the excess CO produced appears stable after melting, and consistent between the sample types, the excess CO observed is likely a systematic extraction system blank.

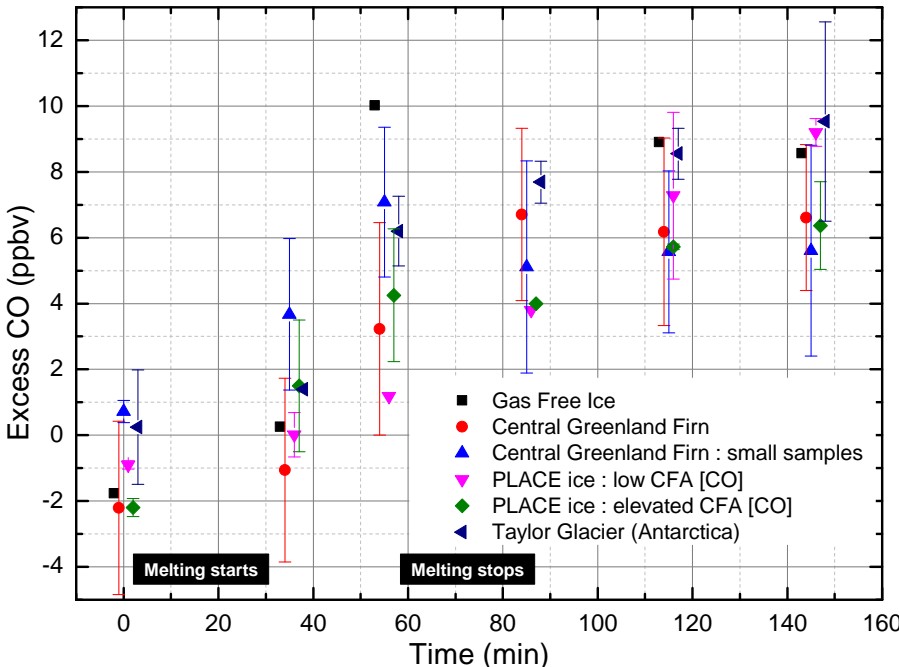

**Figure 3.** Average excess in CO mixing ratios produced during melting of discrete samples measured by gas chromatography combined with a mercuric oxide reduction detector. The CO values shown have been corrected for a small ($\sim$ 3 ppbv) CO excess observed in equivalent tests where the ice samples were not melted (see Sect. 1.9.3 in the SI). The following samples are investigated : gas free ice, firn from Central Greenland (including low mass samples), PLACE ice samples exhibiting high CFA [CO] and low CFA [CO], Taylor Glacier (Antarctica) ice. Samples description is reported in Table S4. Error bars represent one standard deviation for all sample runs of a given type and points without error bars represent single measurements

.





If the CO production mechanism involves organic compounds present in the ice lattice, it would be expected that variations in the concentrations of organics from samples collected at different sites / from different depth levels, as well as differences in the melted ice mass would result in significant differences in excess CO. Some of the PLACE ice samples were specifically
selected for having exhibited low and stable [CO] on the continuous CFA record while others were selected for exhibiting elevated and spiky [CO] behavior suggesting the possibility for instantaneous CO production during melting (Fig. 3, Table S4). The other samples, the two Taylor Glacier and gas-free ice were also selected as they should have very different trace organic loadings to the Greenland ice, especially the gas-free ice sample which have no significant organic loading (< 2 ppbC). Further, replicate firn samples of different mass would be expected to have a total organic loading that scales with the firn mass. Because
there are no significant differences observed in the excess CO produced from these very different sample types (Fig. 3), these results indicate that the melt-extraction process itself does not result in significant CO production from trace organics found within the ice samples. Instead, these results lend further support to the idea that the elevated and highly variable [CO] values observed in Greenland ice samples are due to excess CO produced from in situ production within the ice itself (Faïn et al., 2014; Haan and Raynaud, 1998).

### 3.3 Extracting atmospheric CO from Greenland ice cores

We now explore if past atmospheric concentrations of CO can be extracted from the low frequency variability of the CO ice-records' baselines (Fig. 1). The key question is: does the in situ production present in all five cores still impact the $5^{th}$ percentile of data that we adopt as a baseline?

In our earlier study on the NEEM-S1-2011 core, we reported co-variation between high-resolution CO, and pyrogenic aerosol
such as refractory black carbon (rBC) or ammonium ($NH_4^+$), with abrupt, narrow peaks in all three often (but not always) coinciding (Fain et al., 2014). Within the 50 cm length core analysed in that study, elevated $NH_4^+$ and CO were found together with dissolved organic carbon (DOC) (>100 ppbC) representing an important reservoir of carbon, which could potentially be oxidized into CO. The oxidation processes involved, however, remain unidentified. Here we focus on the two cores for which we have obtained reliable total organic carbon (TOC) data (Sect. 2.4). These are interesting cores to compare because they
have very different snow accumulation rates: Tunu13 has the lowest accumulation rate, which also varies over time between 8 to 13 $\mathrm{cm\,weq\,yr^{-1}}$, while PLACE is more typical of a Late Holocene Greenland core with a stable accumulation rate of $\sim$20 $\mathrm{cm\,weq\,yr^{-1}}$.

#### 3.3.1 The importance of data resolution

Typical CO, TOC, and $NH_4^+$ patterns observed along the PLACE core are reported along a 1.6 m section on Fig. 2. Maximum
and minimum levels of the three species occur at similar depths. $NH_4^+$ is commonly used in Greenland ice to identify summer layers that are characterized by a well-marked summer maximum (Legrand and Mayewski, 1997). Our data indicate that at PLACE both TOC and CO peak simultaneously in ice layers deposited in summer.

Typical CO, TOC, and $NH_4^+$ variations along the Tunu13 core appear different to those at PLACE (Fig. 4). The seasonal cycle in $NH_4^+$ is still detectable but the CO and TOC records show much less co-variation than at PLACE. The higher degree of




smoothing in CO relative to PLACE results in part from Tunu13 being analysed with the 2013 DRI CFA setup (Micromodule degasser), which shows a larger response time in the gas phase (Table S2). The effect is compounded by the much lower snow accumulation rate of Tunu13 relative to PLACE - the section of core in Fig. 4 has only $8\,\mathrm{cm\,weq\,yr^{-1}}$. Step tests (Sect. 2.2.7) suggest that for CO a 60% attenuation would be expected for a sine signal of 10 cm wavelength, a depth interval representing <1 yr in this case. PLACE and Tunu13 TOC were analyzed with the same setup (i.e impacted by the same analytical smoothing). However, and similarly to CO, the lower accumulation rate of Tunu13 results in a larger smoothing of the TOC signal.

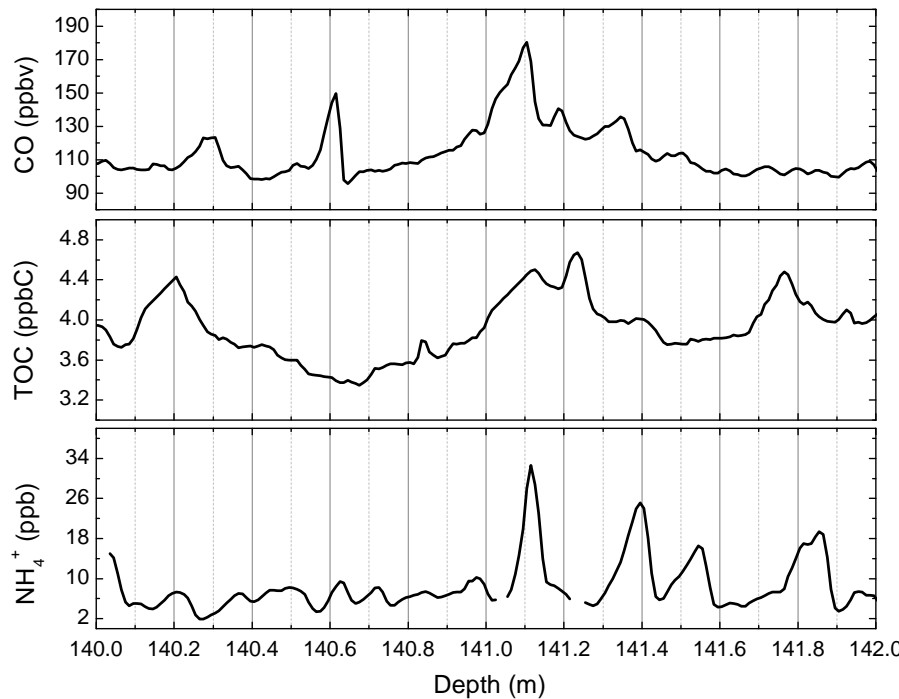

**Figure 4.** High-resolution TOC, NH$_4^+$, CO from the Tunu13 core. Snow accumulation is about $8\,\mathrm{cm\,weq\,yr^{-1}}$ at this depth.


The significant analytical smoothing for the Tunu13 ice-core suggests that annual winter minima in CO (associated with winter minima in TOC) may be unresolvable, which would lead to an overestimation of the baseline even when using the $5^{th}$ percentile values. In contrast, repeated analyses of the higher snow accumulation PLACE core on two different gas CFA setups (DRI and IGE, Table S2) show that despite the DRI setup (2015 configuration) exerting a greater smoothing effect than the IGE one (SI section 1.8, Fig. S10 and S11), the CO minima values are well-resolved in both cases (Fig. 2). Furthermore, the PLACE CO

baseline extracted from the IGE continuous record is similar to the one based on the DRI dataset (SI section 2.4), suggesting that analytical smoothing is not a limitation on isolating CO levels in the winter ice layers at this site. This suggests that the $5^{th}$ percentile baseline calculated at PLACE and other high snow accumulation sites is suitable for investigating past evolutions of the atmospheric CO burden.





### 3.3.2 TOC patterns varying with accumulation rate


The high resolution TOC measurements collected along the PLACE core exhibit a stable TOC background value of 13.5 ppbC (baseline defined as the $5^{th}$ percentile), with 17 spikes above 40 ppbC (70 m long record, Fig. S18). These spikes are often co-located in depth with increases in CO concentration (e.g., Fig. 2) as observed previously on the NEEM-2011-S1 core (Faïn et al., 2014), suggesting that in situ CO production may be related to the organic carbon (OC) availability. TOC levels observed

at Tunu13 are lower, with a baseline value of 7.9 ppbC ($1\sigma$) and only 10 spikes above 40 ppbC for a longer record (143 m long record, Fig. S19). However, mean TOC (10 meters window averaging) exhibits a low frequency variability with values ranging from 4 to 12 ppbC (Fig. 5). This low frequency change is correlated to change of the snow accumulation rate (Fig. 5 which varies from 8 to 13 $\mathrm{cm\,weq\,yr^{-1}}$ (annual average). Along most of the core, this linear relationship between TOC and snow accumulation rate remains unchanged, although it becomes steeper below 160 m depth (Fig. S20). Such correlation is not observed at PLACE where the snow accumulation is constant, with annual mean values ranging 20-21.5 $\mathrm{cm\,weq\,yr^{-1}}$.

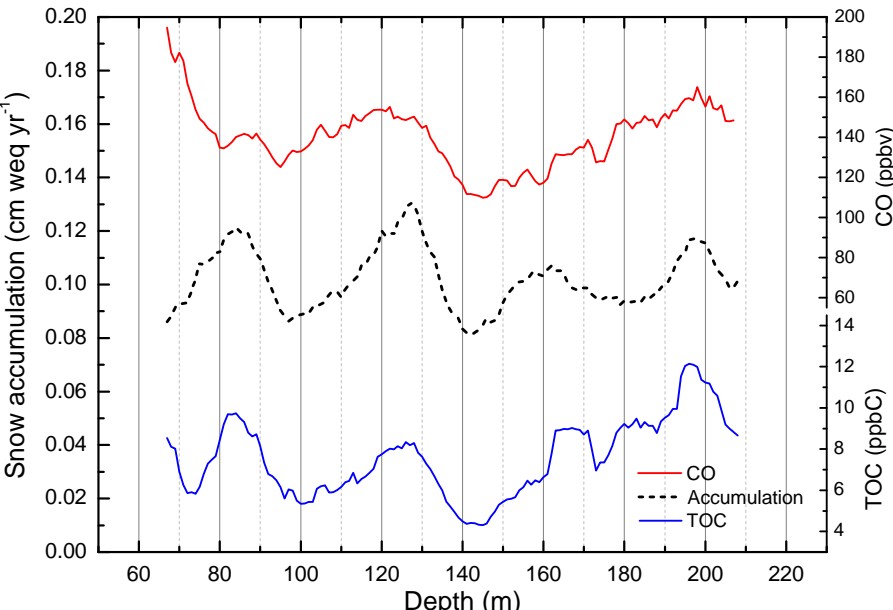

**Figure 5.** Running averages (10 m windows) of TOC and CO concentrations for the Tunu13 core, plotted along with annually averaged surface snow accumulation rate.


Our results indicate that the fate of TOC deposited at Tunu13 is related to the rate of snow accumulation at the surface. Both Tunu13 and PLACE datasets suggest that the lower the accumulation rate, the lower the TOC baseline concentrations in the deep ice. A snowpit study of recent snow layers deposited at Summit (Hagler et al., 2007) showed that a large fraction of water soluble and insoluble organic carbon may be lost at the surface during post-deposition processes, such as photochemical

reactions. We hypothesize that such processes may be enhanced at lower accumulation rates when the snow layers get exposed to UV radiation for longer.





However, such processes may not be the only ones at play, as suggested by investigating the relationships between TOC and ammonium. Forest fire debris reaching Central Greenland snow mainly consists of ammonium formate, and organic carbon is mostly made of formate at Summit (Legrand et al., 2016). Legrand and De Angelis (1996) reported a formate-to-ammonium

molar ratio close to unity in elevated ammonium events recorded at Summit both during the last 200 years and the Holocene. The relationship between TOC and ammonium at PLACE (Fig. S21) exhibits a similar slope close to unity. A completely different pattern is observed at Tunu13 (Fig. S22), with unexpectedly low TOC observed along ammonium peaks. We suggest that ammonium formate has been remobilized after deposition in the snow at sites with snow accumulation rates lower than 12 cm weq yr$^{-1}$. While ammonium remains when deposited due to acidic snow layers, formate evolves to a different form,

presumably gaseous formic acid that, in contrast to ammonium, can escape from neighbouring acidic snow layers and possibly partly be released into the free atmosphere above the snowpack. Such a process could contribute to explain lower TOC levels observed under lower snow accumulation rates, and also the smoothed shape of the TOC peaks at Tunu13 (Fig. 4).

### 3.3.3    TOC patterns and in situ CO production

In situ CO production within Greenland ice archives is very often co-located in ice layers where TOC levels are high. The

importance of organic carbon in the processes driving CO in situ production is supported by the long-term averaged mean CO record (running windows of 10 m) which is significantly correlated to mean TOC at Tunu13 (r$^2$ = 0.56, p<0.01, Fig. 5) for preindustrial times (gas age prior to 1850 CE, i.e. below 83 m depth), with increases in mean CO when TOC and snow accumulation are higher. However, comparison of the Tunu13 and PLACE datasets suggest that higher TOC levels does not always imply higher CO concentrations: over the 1640-1950 CE gas age time period, the mean CO mixing ratio is almost

identical for Tunu13 and PLACE (137 and 136 ppbv, respectively) while mean TOC is higher for PLACE compared to Tunu13 (20 and 7 ppbC, respectively). This may suggest that the required amount of OC to lead to significant in situ CO production is already reached at the Tunu13 site.

While the mean CO mixing ratio is significantly correlated to the mean TOC concentrations in ice at Tunu13, this is not the case for baseline CO level which does not exhibit significant correlations with mean or baseline TOC concentrations. However,

similarities in trends and patterns can be seen when comparing Tunu13 baseline CO and TOC (Fig. 6). The larger analytical smoothing impacting the Tunu13 CO record means that some of the CO baseline signal likely incorporates in situ produced CO from spring/summer ice layers. We can not rule out, however, that a redistribution of organic carbon along depth driven by OC post-deposition process (shifting the ammonium-formate equilibrium with the gas phase) impacts specifically the Tunu13 CO record by providing some additional organic substrates in winter layers.

### 3.3.4    Potential of low snow accumulation sites for long-term CO reconstruction

There are signs that over the long term (i.e., 500 yrs and more), low accumulation sites such as Tunu13 are promising for reconstruction of paleoatmospheric CO records. For example, the deepest section of the Tunu13 CO record does not show an increase in MAD (Sect. 3.1) and exhibits a relatively stable baseline. This is different from higher accumulation sites such as NGRIP or NEEM, where MAD increases with depth, and a positive trend in [CO] baseline with depth is clearly observable





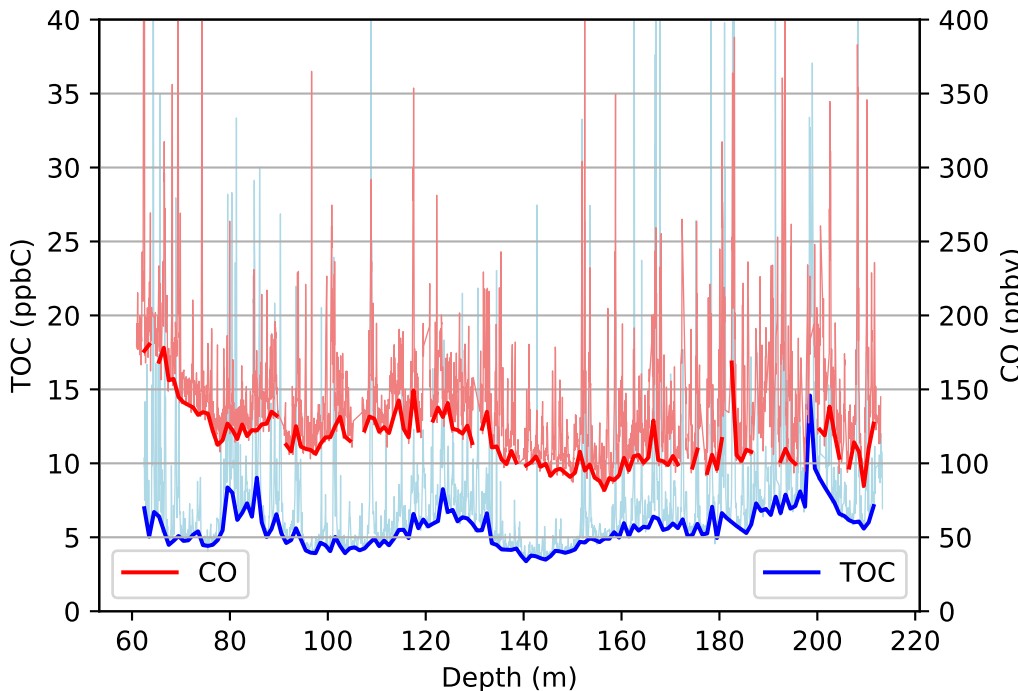

**Figure 6.** Running averages (Continuous (high-resolution) and baseline ($5^{th}$ percentile calculated over a 1 m window) CO (red) and TOC (blue) records for the Tunu13 ice core.

(Fig. S15). The chemical processes involved in CO in situ production are likely multiple and complex and the oxidants involved remain unknown. Nevertheless, we observe that $H_2O_2$, a potential oxidant, exhibits lower levels when accumulation is lower (i.e., Tunu13 vs PLACE, Fig. S23).

### 3.3.5 Summary

Investigating the CO and TOC PLACE records did not reveal any evidence that the CO baseline could be influenced by in
situ production at that site. In contrast, Tunu13 has a lower snow accumulation rate than other sites (Table 1) and thus the analytical CFA smoothing of its CO record is more pronounced. This effect likely introduces a positive bias into the Tunu CO baseline record for ice sections impacted by in situ production. This together with the similarities between baselines TOC and CO observed along the Tunu13 record (Fig. 6) precludes interpretation of the Tunu13 baseline as an atmospheric signal in this study.





### 3.4 Comparison of PLACE and Eurocore datasets

Haan et al. (1996, 1998) published a pioneering CO dataset based on discrete sampling of the Eurocore ice core. This dataset revealed a smoothed CO signal, interpreted as not being affected by in situ production over the last 300 yrs (26 samples, depth range: 78-154 m). However, Haan et al.'s CO data for the period preceding 1700 CE displayed a relatively elevated and scattered signal with levels fluctuating from 90 to 180 ppbv, likely impacted by in situ production. Through our study, we were unable to identify a Greenland site where the ice archive provides a low and stable CO record similar to Haan et al.'s over the last few centuries. Specifically, the CO record retrieved from the PLACE ice-core, which was drilled less than 1 km away from the Eurocore borehole, shows high frequency variability along the entire CFA record (80-153 m depth), and this observation is supported by discrete measurements conducted on the PLACE core with an analytical process in principle similar to the one deployed by Haan et al. (Sect. 2.2.6) (Fig. 7).

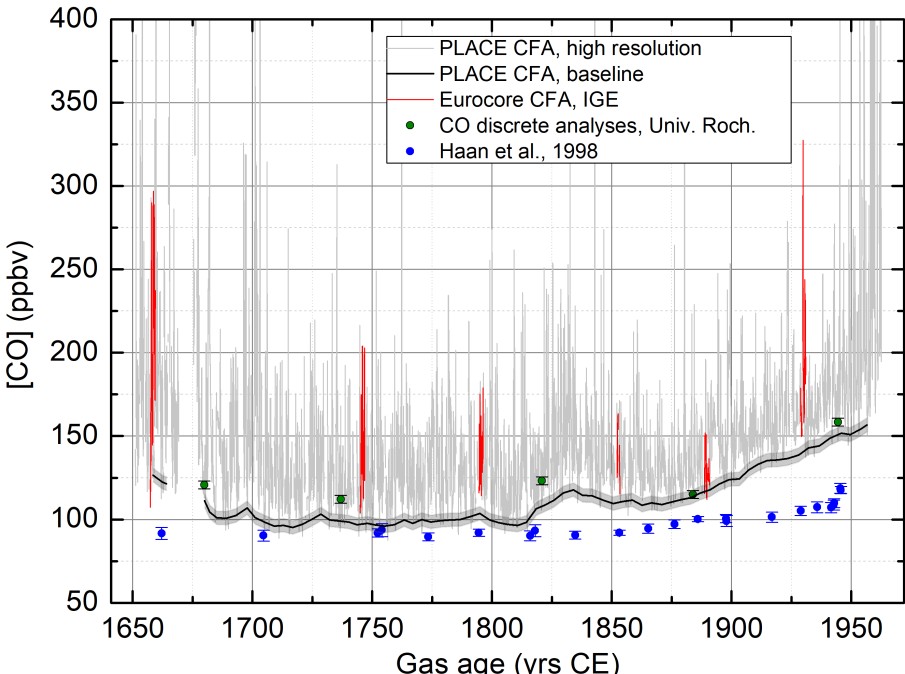

**Figure 7.** CO mixing ratio collected (i) along the Place ice core: continuous IGE CFA record, with $5^{th}$ percentile baseline (black line), and discrete dataset (green dot); (ii) along the Eurocore archive : historical Eurocore discrete CO data (blue dot, Haan et al 1998), and CFA-based dataset measured in 2019 at IGE (red).

Over the 1700-1900 CE period, Eurocore discrete CO data are about 10 ppbv lower than the baseline of the PLACE CFA-based record (Fig. 7). Furthermore, the trend and rate of change of the two records are similar from 1700 to 1905 CE period but diverge over the 1925-1950 CE interval, with an increasing offset between datasets, and an extremely sharp increase in CO concentrations observed only in the Eurocore record for the 1935-1950 CE time period.





To further investigate this apparent offset between our new Greenland CO records and the Eurocore CO record of Haan et al., we analysed six 50 cm-long sections of the 1989 Eurocore ice core with the gas CFA IGE setup in January 2019, at depths spanning 84 to 153 m. A fraction of the Eurocore core has been stored as an archive for 30 yrs at -20°C. We were able to sample the Eurocore archive at exactly the same depths that Haan et al. did in the 1990s (the gaps left by the discrete sampling conducted by Haan and colleagues could be seen). Figure 7 shows Eurocore data (both from Haan et al. and our new CFA dataset), along with the PLACE IGE CFA record: to do so, an offset of +6 m was applied to the Eurocore depth scale, so as to take into account 26 yrs of snow accumulation that occurred between the Eurocore and PLACE drillings. Our CFA-based CO analyses reveal high variability along the Eurocore archive. We systematically observed higher CO mixing ratios than previously reported by (Haan and Raynaud, 1998) including at the exact depths sampled by them. We established that such discrepancies were not related to differences in CO calibration scales by reanalyzing in 2015 the standard gases used by Haan el al. for calibrating their Eurocore dataset in the 90s. Overall, our CFA-based Eurocore data are in contradiction with the historical Eurocore dataset but are in good agreement with the PLACE record, in terms of both absolute levels and high variability. The discrete PLACE CO data, measured with a methodology similar to the one applied by Haan et al., also exhibit higher values than Haan et al. dataset (Fig. 7), while they show an excellent agreement with the CFA CO dataset (Fig. S7).

To summarize, we have not been able to reconcile the historical Eurocore dataset of Haan et al. with a new Greenland ice-core CO dataset (PLACE). This is despite analysis of remaining Eurocore ice from depths co-located with the published Haan data and the fact that PLACE ice core was collected just 1 km away from Eurocore. The reasons for the relatively low and stable CO measurements reported in the 1990s for the Eurocore ice core remain unknown and our results bring the validity of the Haan et al. (1996, 1995) results into question.

## 3.5 Constraining past atmospheric CO in the northern hemisphere

### 3.5.1 CO baseline records over the last 300 yrs

Extracting atmospheric information from a single CO record retrieved from a Greenland ice archive affected by in situ production features is challenging (e.g., Faïn et al., 2014). However, we show here that four records, reconstructed from ice archives originating from different Greenland locations and drilled over a time period spanning more than a decade, exhibit common patterns. Fig. 1 reports $5^{th}$ percentile CO baselines extracted over the last 300 years from the D4, NGRIP, NEEM, and PLACE records (DRI dataset). The four CO baseline records all lie within an envelope of maximum 20 ppbv range. From 1700 to 1875 CE, the records reveal stable or slightly increasing values remaining in the 100-115 ppbv range. From 1875 to 1950 CE, the records indicate a monotonic increase from 100-120 ppbv to 135-150 ppbv (i.e., a mean rate of increase of $\sim 0.3$ ppbv yr$^{-1}$). Overall, Fig. 1 reports a $\sim 30\%$ increase in CO concentration at high latitudes of the northern hemisphere atmosphere from pre-industrial to 1950 CE. Interestingly, in 1835 CE, PLACE, NGRIP, NEEM, and D4 baselines exhibit a common maximum in CO concentration but at a level just slightly higher than concentrations observed during the 1700-1820 CE period.

A multisite composite CO baseline spanning 1700-1957 CE was generated by considering D4, NEEM, NGRIP, and PLACE records and excluding Tunu13. For consistency, we only consider CO records collected on the same CFA gas setup (DRI).





The multisite composite, reported in Fig. 8, was obtained by averaging CO baselines values interpolated on a common gas age scale. An uncertainty envelope (1σ) obtained by considering the independent uncertainty of individual records is plotted along with the composite.

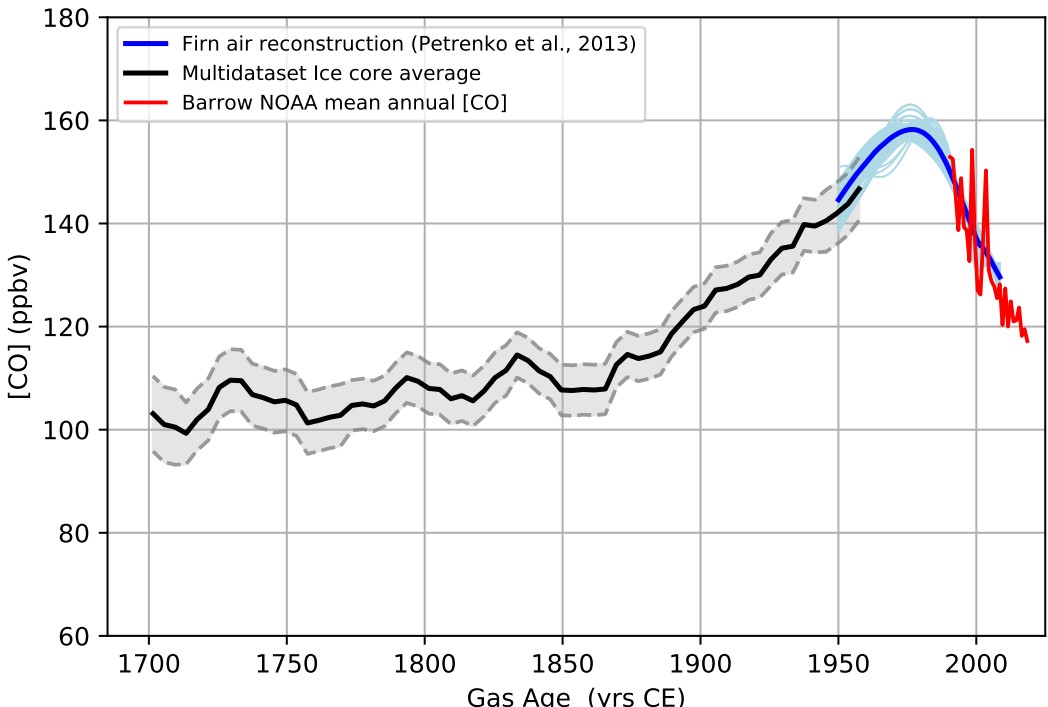

**Figure 8.** Past atmospheric CO mixing ratios for the Northern Hemisphere high latitudes and spanning 1700-2018 CE. Black line : multisite average obtained by combining baselines ($5^{th}$ percentiles) of continuous CO records collected at four Greenland sites (PLACE, NEEM, NGRIP and D4), representing an upper bound estimate of past atmospheric CO burden. Light blue : firn air records obtained by combining samples from NEEM, NGRIP, and Summit; blue : average firn air record (Petrenko et al., 2013) ; Red line : NOAA atmospheric monitoring at Barrow (Alaska, USA).

The fact that the CO baseline records from the four different Greenland ice cores are so consistent gives us confidence that our multisite reconstruction provides a reliable reconstruction of past atmospheric changes. An additional argument for that conclusion comes from the good overlap between our multisite reconstruction and CO data from firn air measurements conducted at Summit, Greenland (see section 3.5.2). However, we cannot exclude the possibility that past atmospheric CO levels could have been lower than shown in Fig. 8 and recommend that our multisite average should be taken as an upper bound of
past atmospheric CO burden. If in situ CO production drives the high frequency variability seen throughout the CFA records of





these four ice cores, as we hypothesize (see Sect. 3.2), it is impossible to exclude the possibility that minimum CO levels may also be slightly affected (Sect 3.3).

### 3.5.2 Comparison with firn air reconstruction

An independent means to assess the validity of our atmospheric CO reconstruction is to compare it with a reconstruction of

northern hemisphere high-latitude atmospheric CO mixing ratios from Greenland firn air, available for the period 1950-2008 CE (Petrenko et al., 2013). This reconstruction includes firn air samples collected at three deep ice core sites in Greenland (NGRIP in 2001, Summit in 2006, and NEEM in 2008), and shows that CO records from these three sites agree well with each other as well as with recent atmospheric measurements from Barrow, Alaska. The reconstructed firn air history suggests that Arctic CO mixing ratio in 1950 CE was 138–148 ppbv, rose by 10–15 ppbv from 1950 to the 1970s, peaked in the 1970s or

early 1980s, and finally declined by about 30 ppbv to present-day levels. The average firn air-based CO history spanning the last six decades (Petrenko et al., 2013) is shown on Fig. 8 along with our ice core CO composite reconstruction. The uncertainty in this scenario is captured by the variability of a set of 61 possible scenarios (Petrenko et al., 2013), and is shown as an envelope on Fig. 8.

PLACE and D4 high resolution CO baseline records overlap with the firn record. The firn air data suggest that CO concentration

spanned 138-148 ppbv in 1950 CE for the Arctic atmosphere Fig. 8). The 1950 CE CO concentration range as defined by PLACE and D4 baselines is 135-150 ppbv, indicating good agreement between firn air and ice core reconstructions.

### 3.5.3 Spatial representativity of the record

To investigate the spatial representativity of the ice core record, we compared the [CO] outputs simulated in the frame of the ACCMIP exercise (Lamarque et al., 2013) and averaged over three different areas: the 45-90°N latitudinal band, an area

encompassing the Greenland ice sheet (20-60°W; 60-84°N), and the single grid point including the PLACE drilling location for the years 1980 and 2000. Average [CO] over Greenland and [CO] at the PLACE drilling location are identical. However, atmospheric [CO] simulated for the 45-90°N latitudinal band is about 12% higher than [CO] simulated over Greenland. These results support our approach to combine ice core reconstructions obtained from different Greenlandic locations, but indicate that the absolute CO concentrations derived from our datasets are representative of the Greenlandic, Arctic, atmosphere, and

may underestimate the average CO concentrations for the entire 45-90°N latitudinal band. On the other hand, the temporal changes depicted by our composite record probably have a larger spatial significance than the Greenlandic context alone.

## 4 Summary and conclusions

New continuous profiles of CO mixing ratio have been measured along five Greenlandic ice cores, the PLACE, NGRIP, D4, Tunu13 and NEEM-SC archives. We also revisited the NEEM-2011-S1 dataset (Faïn et al., 2014). By coupling ice core melter

systems with online measurements (SARA spectrometer), 700 m of ice were analyzed at high resolution for CO mixing ratio. This methodology has greatly improved over the last decade, with excellent external precision, low and well-constrained blank



values, and a good accuracy. All investigated records revealed high and variable concentrations that can not be interpreted as changes in atmospheric CO, but are most likely related to in situ production within the ice archives themselves. Additional measurements conducted on individual discrete samples allowed us to rule out the possibility of rapid CO production from

organics in the ice during melting or other large CFA system artifacts. We have not been able to reconcile the prior Eurocore dataset of Haan et al. (1996, 1998) with any of the new Greenland ice core CO measurements made by both discrete and continuous analytical methods. Our results therefore question the accuracy of the relatively low and stable CO values reported by Haan et al. (1996, 1998) for the Eurocore ice core.

We have presented a multisite average ice core reconstruction of past atmospheric CO for the northern hemisphere high lat-

itudes, covering the period from 1700 to 1950 CE, providing an upper bound estimate of past atmospheric CO burden. This signal of paleo-atmosphere was extracted from the low frequency variability of the CO records' baselines of the PLACE, D4, NEEM, and NGRIP archives. From 1700 to 1875 CE, the multisite average record is stable with a slight increase with time from 103 to 114 ppbv. Our upper-bound value for CO mixing ratio at preindustrial times in the Arctic is 110 ppbv. From 1875 to 1950 CE, the CO increases monotonically from 114 to 142 ppbv, with a rate of increase of about 0.3 ppbv yr$^{-1}$. Finally, an

excellent agreement between our ice core based record and previously published firn air reconstruction is observed in the 1950s (Petrenko et al., 2013). The ice-core and firn [CO] histories, spanning 1700-2010 CE, exhibit similar patterns to an up-to-date combined inventory of anthropogenic and open burning CO emissions (van Marle et al., 2017; Hoesley et al., 1998). Our CFA-based multisite Greenland ice core CO reconstruction provides an indication of the timing and magnitude of past variations in the high northern latitude CO burden that can provide a benchmark for future atmospheric chemistry model studies. Specif-

ically, a natural extension of this work should be the comparison of our ice-core-based atmospheric CO reconstruction with model outputs from the AerChemMIP exercise (Collins et al., 2017).

*Acknowledgements.* This work was supported by the following programs: the French ANR projects RPD-COCLICO (#10-RPDOC-002-01, X.F.) and NEEM (#07-VULN-09-001), the EU grants FP7-IP #ENV-2010/265148 (project Pegasos, X.F.) and FP7 ERC #291062 (project Ice&Lasers, J.C.), the US NSF awards #1406236 (V.V.P.), #0221515 (J.R.M.), #0909541 (J.R.M.), #1204176 (J.R.M.), #1406219

(J.R.M.), and #0968391 (E.B., Partnerships in International Research and Education, project PIRE), the Packard Fellowship for Science and Engineering (V.V.P.), and finally the French national program LEFE/INSU (project GreenCO, X.F.). Grateful thanks go to Olivia Maselli, Larry Layman, Daniel Pasteris, Michael Sigl, and other members of the DRI team who assisted with the measurement campaigns. We thank Frederic Prié, Elise Fourre, and Amaelle Landais for their help in the continuous measurement of water isotopes along the PLACE core at IGE, in 2017. We are grateful to Ray Langenfelds for the measurements of gas standards at CSIRO, in 2015. We thank Sophie Szopa and

Kostas Tsigaridis for our useful discussions. Polar Field Services and the 109$^{th}$ New York Air National Guard, and the French Polar Institute (IPEV) provided logistical support for ice core drilling. We are grateful to drillers and field teams. The NEEM project is directed by the Center for Ice and Climate at the Niels Bohr Institute, Copenhagen, and the US NSF OPP. It is supported by funding agencies and institutions in Belgium (FNRS-CFB and FWO), Canada (NRCan/GSC), China (CAS), Denmark (FIST), France (IPEV, CNRS/INSU, CEA and ANR), Germany (AWI), Iceland (RannIs), Japan (NIPR), Korea (KOPRI), The Netherlands (NWO/ALW), Sweden (VR), Switzerland (SNF),





United Kingdom (NERC), and the USA (US NSF, OPP). NGRIP is a multinational research program funded by participating institutions in Denmark, France, Germany, Japan, Sweden, Switzerland Belgium, Iceland, and the United States.

*Data availability.*   The high resolution carbon monoxide datasets will be made available on the World Data Center for Paleoclimatology.

*Author contributions.*   This scientific project was designed by XF, JC, VVP, RR, EB, and TB. The high resolution carbon monoxide measurements were carried out by XF and RR, with support of KF, TB, JC and EB. Discrete CO analyses were carried out by PP and EC, and

chemistry analyses were carried out by NC and JRM. XF, PP, and VVP participated in the PLACE drilling. RR participated in the Tunu13 drilling. The codes for data processing and modelling were developed by KF, JRM, and XF. All authors contributed to the interpretation of the data. The manuscript was written by XF with the help of all co-authors.

*Competing interests.*   The authors declare having no competing interests





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
