# Peer review of "Northern Hemisphere atmospheric history of carbon monoxide since preindustrial times reconstructed from multiple Greenland ice cores"

_Climate of the Past, 2021_

## Author Comment (AC1)

**Response to Referees – cp-2019-94**

We are thankful to Murat Aydin and Maria Elena Popa for their constructive comments on the article. We listed below our responses to the major and minor specific comments.

The comments of the referees are in blue, and our corresponding responses are below in black. All the section and figure numbering included in our response to reviewers refers to the updated manuscript and SI.

Xavier Faïn on behalf of all co-authors

**Review #1 from Murat Aydin**

For clarity we have sometimes regrouped different comments of the referee, to provide a common answer.

I still have suspicions about the spikes being entirely due to in situ production.

Our data suggest that other potential processes driving the spike can essentially be excluded, as discussed in this response to referees.

I'm not convinced by the arguments for omitting the Tunu record from the final composite product.

The Tunu record has potential, however we conservatively exclude it from further analyses due to (i) a positive bias related to analytical smoothing and in situ CO production, possibly reaching 8 ppbv locally, and (ii) evidences that TOC is remobilized after deposition at Tunu13, and thus possible occurrence of CO in situ production at any depth for this ice core.

I think it is possible all records including the firn may be somewhat elevated above true atmospheric CO levels.

We agree with Murat Aydin and state that our record should be considered as an upper bound of past atmospheric CO burden. We also now mention that in situ production in the firn itself could cause a +5 ppbv contamination to CO measurements (Petrenko et al., 2013).

The manuscript provides some discussions on possible mechanistic explanations for in situ production of CO, but does not make tangible progress.

Our study includes laboratory measurements to investigate the occurrence of CO production during Greenland samples melting. More laboratory tests and protocols would be required to evaluate closely mechanisms driving CO in situ production. However, we believe that sharing hypothetical mechanistic explanations and numerous datasets (CO, and TOC) is already a progress.

From an organizational perspective, it would help if the references to the supplement from the main text specifies exactly which section in the supplement is being referred to.

All references to the supplement included in the main text have been revised accordingly.

**CO spikes in the CFA records**

The reviewer is "struggling to envision where this production happens and over what kind of time frame?". We are struggling similarly, and our paper does not report a fully comprehensive interpretation of the occurrence of elevated, abrupt, CO spikes in Greenland ice core. However, new results included in our manuscript clearly extend our understanding of abrupt CO patterns, notably by including (i) continuous CO records from five different sites, and (ii)) specific tests conducted on discrete samples (Sect. 3.2.4), (iii) for the first time continuous, high resolution, TOC dataset.

We understand the reviewer's statement "To me, the term in situ production implies the type of observations from the deeper sections of NEEM and NGRIP ice cores: the amplitudes of the peaks and the mean levels grow with depth due to slow but continual CO production over long periods". However, we are still lacking fundamental knowledge about the in situ CO production process to conclude that patterns as described by the reviewer should apply to all records/sites. In our study, the Tunu13 record exhibits a limited increase in CO spike amplitude with ice age (considering that the D4 record is too short for a comparison with others records). We can not exclude that lower accumulation at the Tunu13 site would drive different chemical processes within the ice matrix, impacting CO.

Can this happen without a firn component? The smoothed firn air records (Petrenko et al., 2013) observed at different Greenland sites indeed contrast with the elevated CO values observed in CO spikes. Firn air sampling devices collect air from layers mainly where open porosity exists. We can not exclude that in situ CO production already occurs in deep firn layers where porosity is already closed. However, we were not able to evaluate such a hypothesis. CO CFA data can not be collected in the deep firn where bubbles close (within the Lock In Zone), because lab air infiltrations through open porosity occur during the measurement process. No solid CFA record was available either just below the LIZ because of post coring air entrapments (impacting 5 meters down below the close-off at PLACE, see Fig. R2 included in this response). Finally, no evidence for higher [CO] in early trapped layers revealed by negative layering CH4 artefact could be observe; this actually makes sense because layering is first related to firn physical properties, while CO in situ production is more likely related to ice matrix chemical composition (e.g., TOC levels), the 2 factors are not coinciding.

However, we have modified our manuscript (Sect. 3.5.2) to include a potential +5 ppbv contribution of CO in situ production at the bottom of the firn as stated by Petrenko et al. (2013).

**Fig. 7: It would be useful to see the mean of the PLACE CFA record in this figure.**

**Note : Fig. 7 is now Fig. 6 in the revised manuscript. It is referred to Fig. 6 below.**

The PLACE samples which were selected for discrete analyses were all selected within depth intervals where the continuous CFA CO record previously analyzed revealed low, possibly atmospherically relevant, values. For this reason we choose in the initial submission to report on Fig. 6 the CFA CO baseline signal.

We understand that this choice is confusing, and indeed [CO] from discrete samples should be compared with mean CFA [CO] calculated over the same depth intervals (i.e., the ~20cm depth intervals spanned by discrete samples) : this is what we already reported on Fig. S7. A comparison between discrete and CFA CO dataset, based on the depth ranges of discrete samples, can not be plotted on Fig. 6 which shows the full PLACE CFA CO record. Figure R1 shows an example of such comparisons, which requires a focus on the depth scale.

**Figure R1.** A comparison of a discrete CO analysis with the continuous CO CFA record, along the PLACE ice core. The x-error bar reported on the discrete sample shows the top and bottom depths of the discrete sample.

For clarity of the manuscrit, we have removed the CO baseline from Fig. 6, but we did not include the mean CFA CO signal either. Considering the length of manuscrit, we have not included Fig. R1 in SI.

Fig. S7: How were the CFA-based CO concentration corresponding to the discrete measurements calculated? This can be stated in the text or the caption

On Fig. S7, the CFA-based CO concentrations were calculated as the mean CFA [CO] values observed on the depth intervals covered by discrete samples. E.g., the ~132.7 m depth discrete

sample (Fig. R1) actually spans the depth interval 132.59- 132.79 m. The corresponding CFA value was obtained by averaging CFA [CO] on the same depth interval. The caption of Fig. S7 was modified accordingly.

Figs. S16, S17: The comparison of IGE vs. DRI CFA data suggests the spikes may not entirely be related to in situ production. There is higher variance in the IGE record compared with the DRI one. This is especially evident above 110-120 m, resulting in the picture we see in Fig. S17, which shows baselines in much better agreement than mean levels above 115 m. Clearly this is not just a smoothing issue. So, there appears to be some contribution to the spikes from either the analytical method at IGE or something happening during the sample prep stage at the IGE, or what else? Post-coring entrapment is mentioned as a possible mechanism. Is there a way this can happen in one lab vs. the other, or more in one lab vs. the other? What are the respective ambient temperatures in the freezers where the melts are conducted? Are there corresponding spikes, or even mild elevations, in the CFA methane data that may suggest entrapment of lab/modern air?

Figure S16 (resp. S17) is now Fig. S19 (resp. S20) in the revised manuscript

We agree with the reviewer that the comparison of the DRI and IGE CO datasets is puzzling, as the two records exhibit different CO spike patterns. The observations of the reviewer are correct : there is higher variance in the IGE CO record.

Following the reviewer's suggestion, we have investigated closely the relationship between CH4 and CO mixing ratio in the shallow PLACE ice. We observed [CH4] enhancement in the 80-85m depth range in the IGE record compared to the DRI dataset (Fig. R2). This result evidences post-coring entrapment in the 80-85 m depth range, a depth interval exhibiting the most elevated and abrupt CO spikes in the IGE record. The SI was modified to report this finding, and we now restrict the impact of post-coring entrapment on the IGE continuous CO record to the 80-85 m depth interval. Figure R2 is now included in the SI (Fig. S21).

---

## Author Comment (AC2)

**Response to Referees – cp-2019-94**

We are thankful to Murat Aydin and Maria Elena Popa for their constructive comments on the article. We listed below our responses to the major and minor specific comments.
The comments of the referees are in blue, and our corresponding responses are below in black.
All the section and figure numbering included in our response to reviewers refers to the updated manuscript and SI.

Xavier Faïn on behalf of all co-authors

**Review #1 from Murat Aydin**

*For clarity we have sometimes regrouped different comments of the referee, to provide a common answer.*

I still have suspicions about the spikes being entirely due to in situ production.
Our data suggest that other potential processes driving the spike can essentially be excluded, as discussed in this response to referees.
I'm not convinced by the arguments for omitting the Tunu record from the final composite product.
The Tunu record has potential, however we conservatively exclude it from further analyses due to (i) a positive bias related to analytical smoothing and in situ CO production, possibly reaching 8 ppbv locally, and (ii) evidences that TOC is remobilized after deposition at Tunu13, and thus possible occurrence of CO in situ production at any depth for this ice core.
I think it is possible all records including the firn may be somewhat elevated above true atmospheric CO levels.
We agree with Murat Aydin and state that our record should be considered as an upper bound of past atmospheric CO burden. We also now mention that in situ production in the firn itself could cause a +5 ppbv contamination to CO measurements (Petrenko et al., 2013).
The manuscript provides some discussions on possible mechanistic explanations for in situ production of CO, but does not make tangible progress.
Our study includes laboratory measurements to investigate the occurrence of CO production during Greenland samples melting. More laboratory tests and protocols would be required to evaluate closely mechanisms driving CO in situ production. However, we believe that sharing hypothetical mechanistic explanations and numerous datasets (CO, and TOC) is already a progress.
From an organizational perspective, it would help if the references to the supplement from the main text specifies exactly which section in the supplement is being referred to.
All references to the supplement included in the main text have been revised accordingly.

**CO spikes in the CFA records**

The reviewer is "struggling to envision where this production happens and over what kind of time frame?". We are struggling similarly, and our paper does not report a fully comprehensive interpretation of the occurrence of elevated, abrupt, CO spikes in Greenland ice core. However, new results included in our manuscript clearly extend our understanding of abrupt CO patterns, notably by including (i) continuous CO records from five different sites, and (ii)) specific tests conducted on discrete samples (Sect. 3.2.4), (iii) for the first time continuous, high resolution, TOC dataset.

We understand the reviewer's statement "To me, the term in situ production implies the type of observations from the deeper sections of NEEM and NGRIP ice cores: the amplitudes of the peaks and the mean levels grow with depth due to slow but continual CO production over long periods". However, we are still lacking fundamental knowledge about the in situ CO production process to conclude that patterns as described by the reviewer should apply to all records/sites. In our study, the Tunu13 record exhibits a limited increase in CO spike amplitude with ice age (considering that the D4 record is too short for a comparison with others records). We can not exclude that lower accumulation at the Tunu13 site would drive different chemical processes within the ice matrix, impacting CO.

Can this happen without a firn component? The smoothed firn air records (Petrenko et al., 2013) observed at different Greenland sites indeed contrast with the elevated CO values observed in CO spikes. Firn air sampling devices collect air from layers mainly where open porosity exists. We can not exclude that in situ CO production already occurs in deep firn layers where porosity is already closed. However, we were not able to evaluate such a hypothesis. CO CFA data can not be collected in the deep firn where bubbles close (within the Lock In Zone), because lab air infiltrations through open porosity occur during the measurement process. No solid CFA record was available either just below the LIZ because of post coring air entrapments (impacting 5 meters down below the close-off at PLACE, see Fig. R2 included in this response). Finally, no evidence for higher [CO] in early trapped layers revealed by negative layering $CH_4$ artefact could be observe; this actually makes sense because layering is first related to firn physical properties, while CO in situ production is more likely related to ice matrix chemical composition (e.g., TOC levels), the 2 factors are not coinciding.

However, we have modified our manuscript (Sect. 3.5.2) to include a potential +5 ppbv contribution of CO in situ production at the bottom of the firn as stated by Petrenko et al. (2013).

Fig. 7: It would be useful to see the mean of the PLACE CFA record in this figure.
*Note : Fig. 7 is now Fig. 6 in the revised manuscript. It is referred to Fig. 6 below.*
The PLACE samples which were selected for discrete analyses were all selected within depth intervals where the continuous CFA CO record previously analyzed revealed low, possibly atmospherically relevant, values. For this reason we choose in the initial submission to report on Fig. 6 the CFA CO baseline signal.

We understand that this choice is confusing, and indeed [CO] from discrete samples should be compared with mean CFA [CO] calculated over the same depth intervals (i.e., the ~20cm depth intervals spanned by discrete samples) : this is what we already reported on Fig. S7. A comparison between discrete and CFA CO dataset, based on the depth ranges of discrete samples, can not be plotted on Fig. 6 which shows the full PLACE CFA CO record. Figure R1 shows an example of such comparisons, which requires a focus on the depth scale.

[Figure]

**Figure R1.** A comparison of a discrete CO analysis with the continuous CO CFA record, along the PLACE ice core. The x-error bar reported on the discrete sample shows the top and bottom depths of the discrete sample.

For clarity of the manuscrit, we have removed the CO baseline from Fig. 6, but we did not include the mean CFA CO signal either. Considering the length of manuscrit, we have not included Fig. R1 in SI.

Fig. S7: How were the CFA-based CO concentration corresponding to the discrete measurements calculated? This can be stated in the text or the caption
On Fig. S7, the CFA-based CO concentrations were calculated as the mean CFA [CO] values observed on the depth intervals covered by discrete samples. E.g., the ~132.7 m depth discrete

sample (Fig. R1) actually spans the depth interval 132.59- 132.79 m. The corresponding CFA value was obtained by averaging CFA [CO] on the same depth interval.

The caption of Fig. S7 was modified accordingly.

Figs. S16, S17: The comparison of IGE vs. DRI CFA data suggests the spikes may not entirely be related to in situ production. There is higher variance in the IGE record compared with the DRI one. This is especially evident above 110-120 m, resulting in the picture we see in Fig. S17, which shows baselines in much better agreement than mean levels above 115 m. Clearly this is not just a smoothing issue. So, there appears to be some contribution to the spikes from either the analytical method at IGE or something happening during the sample prep stage at the IGE, or what else? Post-coring entrapment is mentioned as a possible mechanism. Is there a way this can happen in one lab vs. the other, or more in one lab vs. the other? What are the respective ambient temperatures in the freezers where the melts are conducted? Are there corresponding spikes, or even mild elevations, in the CFA methane data that may suggest entrapment of lab/modern air?

*Figure S16 (resp. S17) is now Fig. S19 (resp. S20) in the revised manuscript*

We agree with the reviewer that the comparison of the DRI and IGE CO datasets is puzzling, as the two records exhibit different CO spike patterns. The observations of the reviewer are correct : there is higher variance in the IGE CO record.

Following the reviewer's suggestion, we have investigated closely the relationship between CH4 and CO mixing ratio in the shallow PLACE ice. We observed [CH4] enhancement in the 80-85m depth range in the IGE record compared to the DRI dataset (Fig. R2). This result evidences post-coring entrapment in the 80-85 m depth range, a depth interval exhibiting the most elevated and abrupt CO spikes in the IGE record. The SI was modified to report this finding, and we now restrict the impact of post-coring entrapment on the IGE continuous CO record to the 80-85 m depth interval. Figure R2 is now included in the SI (Fig. S21).

[Figure]

**Figure R2.** High resolution, continuous, methane measurements collected along with CO with the DRI (resp. IGE) CFA setup, and reported in blue (resp. In red). The elevated and spiky methane pattern observed on the IGE dataset reveals contamination of the gas IGE CFA by modern air on the 80-85 m depth interval. Post coring entrapment of modern air may be involved.

As already stated in the SI, "Replicate datasets (Fig. S3) further rule out analytical drifts of the DRI or IGE CO baselines during the measurements: in both laboratories, we first melted the entire core, and later some replicate sections were reanalyzed. Such replicate measurements exhibit an excellent agreement with the main records (Fig. S3)." Higher spike values are expected along the IGE CO record, because the analytical CFA smoothing is lower at IGE compared to DRI, and CO spikes are thus better resolved at IGE.

We have no further suggestions for discussing the differences in CO levels observed between DRI and IGE records. DRI and IGE freezers were maintained at similar temperatures (about -20°C). However, mean CFA [CO] collected 18 month apart from two CFA systems located on different continents (i.e., DRI and IGE CFA setups, North America vs Europe) agree within their uncertainty envelopes for the 85-153 m depth interval. While this is already a good result paving the way to improvement in the frame of further studies, it also suggests that the reasons for

differences between IGE and DRI mean CFA [CO] rely on the quantified uncertainties (e.g., calibration, ...).
The text of the Sect. 2.4 of SI was thus modified to highlight this overlap of the uncertainty envelopes of the means [CO] IGE and DRI CFA records.

Finally, the difference between continuous CO records collected at DRI and IGE remains small, with a gap in mean levels never exceeding 10 ppbv for ice sections which are not near the firn-ice transition (i.e., below 85 m depth). Our interpretations on past atmospheric CO burden are based only on 5th baselines extracted from CFA CO records. Thus, the question raised here, although relevant, does not impact the conclusions of our study. This is stated in SI.

**Interpretation of baseline CO as an atmospheric record**
The paper needs more nuance in discussion of the final composite record. Specifically, the possibility that the entire ice core record and the bottom of the firn being impacted by some level of production should be mentioned.
Our manuscript reports a multisite average obtained by combining baselines (5th percentiles) of continuous CO records collected at four sites (Tunu13 excluded), shown in Fig. 7. The manuscript clearly states that this multi-site average is an upper bound estimate of past atmospheric CO burden, notably because we can not fully exclude that in situ production would impact the minimum CO values observed along CO records.
Specifically, this is stated :
-page 1, abstract, line 15
-page 3, introduction, line 85
-page 22, Fig. 7 caption.
-page 23, Sect. 3.5.1, lines 513-515
-page 24, conclusion line 558

We have no new data to discuss the possibility that the bottom of the firn would be impacted by in situ production. Petrenko et al. (2013) discussed this topic in detail by comparing CO firn records from three sites (Summit, NEEM, NGRIP, Section 3.3 of their Climate of the Past paper). They conclude that the firn air [CO] signals are overall representative of the Greenland land surface [CO] history, although they could not completely rule out a small contribution from in situ CO production (up to +5 ppbv). We have modified our manuscript (Sect. 3.5.2) to include this potential, low, contribution of CO in situ production at the bottom of the firn.

**Omitting the Tunu record from the final composite product**
One thing I struggled with is the decision to not include the Tunu record in the final compilation. The first argument for rejecting the Tunu record has to do with analytical smoothing due to a combination of lower accumulation rate at the site and the lower resolution CFA system at DRI relative to the IGE system. The argument makes sense qualitatively, but how large can this effect realistically be?

We have included a new section to SI (1.9.2) where we attempt to quantify the positive bias of the CO baseline produced by the combination of accumulation rate and analytical smoothing. In this section, we present a model where a synthetic CO signal is convoluted with an analytical CFA green function (also read responses to specific comments for more details). This model suggests that for the Tunu13 record, positive bias of up to 8 ppbv can impact the 5th baseline along multi-meters depth intervals. On the other hand, we do not find bias larger than ~2 ppbv for other records.

Fig. S17 shows the difference between mean and baseline for the PLACE records is about 20 ppb. I would expect the impact of analytical smoothing on the baseline at Tunu would be much smaller. Peak attenuation percentage does not translate into the same percentage elevation in baseline.

This is correct, as stated before the order of magnitude of the impact of analytical smoothing on the Tunu13 CO baseline is in the order of magnitude of ~10 ppbv.

More importantly, the Tunu record doesn't really look biased high from the other records in Fig. 1 (hard to evaluate the full record in Fig. S15), except for a brief period around 1810-1820 and again around 1710 (is the accumulation rate lower during these periods?).

Since I do not see the Tunu record as an obvious outlier in Fig. 1, I consider the arguments about why in situ production could impact Tunu baseline but not the other four records to be of secondary importance. If these were first order processes, I would expect the Tunu record to be clearly offset from the others.

We identified two specific anomalies impacting the Tunu13 CO baseline record :

- A positive bias related to analytical smoothing and in situ CO production, possibly reaching 8 ppbv locally.
- Evidences that TOC is remobilized after deposition, and thus possible occurrence of CO in situ production at any depth of the Tunu13 core. How this impact CO baseline could not be evaluated.

These two anomalies may have limited impact, but they do exist. We chose, conservatively, to keep excluding the Tunu13 CO record from the multisite average shown in Fig. 7. However, this is a methodological and rigorous choice, as including the Tunu13 CO record in the multisite average does not lead to different paleo-atmospheric interpretations. Figure R3 compares multisite CO averages, including, and not including the Tunu13 record. Figure R4 compares the Tunu13 CO baseline with the multisite CO average reported on Fig. 7 (i.e., calculated with only the PLACE, NEEM, NGRIP, and D4 records).

[Figure]

**Figure R3.** Past atmospheric CO mixing ratios for the Northern Hemisphere high latitudes and spanning 1700-2018 CE. Black line : multisite average obtained by combining baselines 5th percentiles of continuous CO records collected at four Greenland sites (PLACE, NEEM, NGRIP and D4). Purple line : multisite average obtained by combining baselines 5th percentiles of continuous CO records collected at five Greenland sites (PLACE, NEEM, NGRIP D4, and Tunu13). Light blue : firn air records obtained by combining samples from NEEM, NGRIP, and Summit; blue : average firn air record (Petrenko et al., 2013) ; Red line : NOAA atmospheric monitoring at Barrow (Alaska, USA).

[Figure]

**Figure R4.** Past atmospheric CO mixing ratios for the Northern Hemisphere high latitudes and spanning 1700-2018 CE. Black line : multisite average obtained by combining baselines 5th percentiles of continuous CO records collected at four Greenland sites (PLACE, NEEM, NGRIP and D4). Green line : Tunu13 CO baseline record. Light blue : firn air records obtained by combining samples from NEEM, NGRIP, and Summit; blue : average firn air record (Petrenko et al., 2013) ; Red line : NOAA atmospheric monitoring at Barrow (Alaska, USA).

Finally, I don't understand why the Tunu baseline in Fig. 1 bottom panel is not extended all the way through 1960. The large peak around 1930 can be treated as another missing section if that is the reason. Is that large peak related to an analytical issue?

We realize that a section was missing in the manuscript submitted (this happened when the LateX manuscript was prepared). This section (3.1.1) is entitled "CO baseline levels". It explains why and how we calculate baselines as 5th percentiles. Notably, we report in this 3.1.1 section that CO baseline levels were calculated as the 5th percentile of data every 4 years over a moving window of 15 yrs, and that "5th percentiles were not computed when more than 50% of the data were missing within the windows". For each 15-yrs window, we combined all intervals of missing data larger than 1-months to establish a global window-percentage of missing data.

The reviewer helped us to realize that our definition of missing data interval should be scaled with the accumulation rate of the core. Considering that Tunu13 has a lower accumulation

(about half of the PLACE, NGRIP, or NEEM accumulation rates), we increased our criteria for detection of missing data to 2-months. This resulted in an updated Tunu13 baseline record extending to 1941. We thus updated Fig. 1, and used this updated Tunu13 CO baseline in Fig. R3 and R4.

**A Section 3.3 too complicated**

The last paragraph of section 3.3.2 and related figures about the relationship between TOC and ammonium does not contribute much to the paper in its current form. A clear explanation of how this all relates to in situ CO production is necessary, or it can be significantly reduced or mostly omitted.

We have shortened Section 3.3 of our manuscript to make it more focused and clear. Notably :
- Sections 3.3.2 and 3.3.3 have been merged in a single section now entitled " TOC patterns and in situ CO production".
- Figure 6, which was showing Tunu13 CO and TOC baselines was removed.
- The section entitled "Potential of low snow accumulation sites for long-term CO reconstruction" was removed, as it did not contribute strongly to the manuscript.

Section 3.3.4 is also problematic. The main point of this subsection is that the long term record from Tunu can yield an atmospheric record. This directly contradicts with many of the previous arguments used to disqualify the Tunu record from the interpretation of the last 300 years. I already stated above that I do not agree with the logic to reject the Tunu data from the short term interpretation, so I actually don't have any major issues with what is being said here, although it would be nice to see the actual plots of MAD for NEEM and Tunu.

For clarity of the Section 3.3, we have removed the sub-section 3.3.4. It was not essential to our manuscript.

However, this section was only stating that Tunu had a "potential" for long term [CO] paelo-reconstruction. To use such potential, one would need (i) to improve CFA analytical resolution, and (ii) to better understand the fate of TOC and how it relates to CO in situ production at low accumulation sites. To our opinion, this is not in contradiction with other arguments reported in the manuscript.

A plot of MAD data for NEEM and Tunu13 has been included in SI (Fig. S18).

The last sentence about peroxide and the associated supplemental figures, while interesting, are tangential and can be omitted. TOC, ammonium, peroxide, and how this all relates to CO production in the ice sheet could perhaps be addressed in a separate paper that focuses on possible chemical mechanisms.

For clarity of the manuscript, we have removed all discussion related to H2O2. However, we have kept the discussion related to the ammonium/TOC relationship as it seemed highly relevant to us. This is one of the new pieces of information about CO in situ production processes reported in our study.

We have removed the discussion comparing baselines TOC and CO, and Fig. 6. Indeed, we reported that "similarities in trends and patterns can be seen when comparing Tunu13 baseline CO and TOC (Fig. 6)". This statement was subjective.

**Discrepancy with the previously published discrete Eurocore data.**

The only thing I can think about is the system blank applied to the Haan and Raynaud measurements. Is it known if the blank correction was higher for shallower sections?

Haan et al., (1996) reports a null system blank, i.e. no contamination at all during the melting extraction. This is also described in Haan's PhD dissertation (written in French). So it seems that a zero blank value was considered for the entire EUROCORE CO dataset.

Another thing that comes to mind is to check if there is any long term increasing trends in terrestrial dust or sea salt-related proxies over the depth ranges corresponding to 1825 onward from PLACE. The running hypothesis has been that any in situ effects on CO should be associated with TOC. However, if TOC is not conservative, meaning if the measured TOC levels have been heavily altered by post-depositional processes, it might be more useful to think in terms more conservative proxies of ice impurities.

A 1825 gas age would correspond to 1630 Ice age. We have not found any long term increasing trends in terrestrial dust or sea-salt related proxies over the depth range corresponding to 1630 ice age onward. Although the chemical CFA dataset collected along the PLACE core at DRI was extensive.

One can also note that PLACE and Eurocore ice cores were drilled only 1 km apart, with the goal to get two cores of similar chemical composition.

**Specific comments:**

Line 14: Where does the 20 cm/y comes from?

The abstract reads : "Evaluation of signal resolution and co-investigation of high-resolution records of CO and TOC show that past atmospheric CO variations can be extracted from the records' baselines at four sites with accumulation rates higher than 20 cm water equivalent per year". Thus 20 cm weq yr-1 is the minimum accumulation for the sites used for paleo-atmospheric interpretation in our study (i.e., PLACE, NEEM, NGRIP and D4).

L44: Would be helpful to state the mean background CO.

Novelli et al. (1998) reports atmospheric CO dataset from 49 monitoring stations. They discuss seasonal and spatial patterns in CO mixing ratios that would be too long to report with details in the introduction. Later in the manuscript we include atmospheric monitoring CO data from Barrow station (AK, USA), and we discuss more closely this trend which relates more directly to our ice core CO reconstruction.

L71: CH4 typo.
The manuscript was corrected.

L85-90: The last three sentences may be better suited for the end of the paper, in section 3.5.3 or conclusions.
We kept these sentences in the introduction as it is important for us to state early that this paper will not include atmospheric modelling and interpretation. Another paper (in prep.) will be dedicated to this.

L201-204: How is the relationship between CH4 solubility and CO solubility quantified?
We report that CFA setups are dynamic systems, not at solubility equilibrium (see Sect. SI 1.8). Thus, no theoretical relationship could be extracted to relate CH4 and CO solubility losses. However, we hypothesize that CO and methane dissolution follow the same physical laws: consequently, if a calibration loop is able to reproduce methane preferential dissolution, it should also reproduce CO losses related to dissolution (page 8, line 204).
We extract, for each analytical campaign and each ice core analyzed, CO solubility correction factor from calibration loops experiments which reproduce the CH4 solubility losses. The CH4 solubility losses are well known because discrete, well calibrated, dataset were available.
To clarify this, a sentence was added to the manuscrit, and more details are available in SI Sect. 1.8.

L219-220: Incomplete sentence.
The manuscript was corrected.

L239-240: Are these tests a good analogue for melting in the CFA system?
During CFA analyses, the melting is produced by a contact of the ice and a hot plate (the "melter"). During the discrete analyses conducted at Univ. Roch, samples are melted by applying to the ice melting vessel a warm water bath maintained at 50°C using an immersion heater with built-in circulator.
These melting processes are quite different, but in both cases, the temperature of the liquid phase is kept low during the melting. This is obvious for the discrete melting setup, where liquid and ice phase co-exist. For the CFA, the temperature of the liquid phase is usually below 10°C at the outlet of the melter.
The CFA melting process is really fast, when the discrete melting process takes up to 50 min. However, our results strongly suggest that CO in extractu production is not related to melting, and that length of the melting process is not driving higher CO production.

L285-290: The comment about not comparing the amplitude of variability between D4 and Tunu due to concerns about smoothing is undermined by the later sentence that states all cores

exhibit similar MAD values for the 1700-1950 period. I'm not sure why the sentence about not comparing D4 to Tunu is needed.
The D4 MAD values are indeed larger. This is now stated in the manuscript and shown by Fig. S18.
The sentence "Therefore, in this study we do not directly compare the amplitude of CO variability between D4 and Tunu13, which exhibit a factor of four difference in snow accumulation rate." was not relevant and was removed from the manuscript.

L310-313: The validity of the Haan and Raynaud data from shallower depths is questioned later in the manuscript. Haan and Raynaud also use larger samples. The reference to Haan and Raynaud work here can be omitted.
Haan and Raynaud use ~50 g samples, with samples of sections similar to CFA sticks. However, Section 3.2.1. discusses how ice core drilling could be related to abrupt CO spikes observed in Greenland ice cores, i.e., before sample cutting.
We chose to keep in Sect. 3.2.1 the following sentence : " In contrast, no specific care was taken during the historical Eurocore drilling when handling cores, with respect to sunlight exposure". Indeed, this piece of information comes from one of the co-authors of the study (M. Legrand, who was in the field during the Eurocore drilling) and is not available elsewhere (i.e., it has not been described by Haan and colleagues). One could hypothesize that the lower CO values observed along the Eurocore shallow ice (Haan et al., 1996) could be related to specific care taken during drilling, but we report that was not the case.

Fig. 3: Perhaps change the x-axis label to "Time after melt starts".
The figure was updated.

L355: Was this a significant covariation or not? Often but not always is uninformative. It is never the case that all peaks match anyway.
We have modified the manuscript as follow to report with more accuracy the findings from Faïn et al. (2014): "In our earlier study on the NEEM-S1-2011 core (Faïn et al., 2014), we reported that 68% of the CO spikes were observed in ice layers enriched with pyrogenic aerosols (i.e., exhibiting ammonium levels above 18 ppb)."

L373: Shouldn't it be >1 yr?
This is correct, the manuscript was modified.

L368: The argument here is based on a comparison of Fig. 2 to Fig. 4. These figures show ~2 m sections. Is there a way to support this argument with a method that utilizes to the entire cores?

L387-389: We can all understand something different from "often." Is it not possible to quantify this? I can suggest subtracting the mean or baseline then looking at r2 in a linear regression analysis, for example.

L414: Reemphasizing the co-location of CO spikes and TOC... It's just hard to really buy into it without an analysis of at least one of the data sets in its entirety.

The three comments above relate about how the relationship between TOC and CO is discussed for the PLACE core. We indeed stated in the manuscript that both records are related, but keep this description qualitative and only showed data from a 2-meters section.

Our 2-m plot (Fig. 2) is useful as it allows the reader to visually observe the co-location in depth of CO and TOC spikes for the PLACE core. Here, we only report colocation in depth, and not correlation between TOC and CO concentrations : in other words, while peaks are collocated, their amplitudes vary independently (investigating linear regression as suggested by the reviewer is not conclusive).

To provide a full record comparison of PLACE TOC and CO, we now report in SI distributions in TOC and CO concentrations observed within summer (respectively winter) layers. We define summer as April to September, and winter as October to March. Depth scale uncertainties and analytical smoothing would not allow to investigate seasonal trends with more details. Seasonal TOC and CO distributions are reported in Fig. R5 and R6. Both distributions show systematically lower [CO] (resp. [TOC]) during winter, suggesting that both species exhibit a coherent seasonal cycle along the entire PLACE ice core.

The manuscript was also modified to include these results.

[Figure]

**Figure R5.** Distribution of seasonal CO mixing ratio for the full PLACE record. Winter period (resp. Summer period) is defined as April-September (resp. October-March).

[Figure]

**Figure R6.** Distribution of seasonal TOC mixing ratio for the full PLACE record. Winter period (resp. Summer period) is defined as April-September (resp. October-March).

L397-398: This sentence indicates there is a correlation between accumulation rate and TOC baseline at PLACE. This contradicts the sentence on L395, which states there is no correlation between accumulation rate and TOC at PLACE.

We first compare TOC and accumulation rate for the PLACE core only : no correlation is found, but this is expected as accumulation rate is constant. This is explained as follow : "This linear relationship between TOC and snow accumulation [...] is not observed at PLACE where the snow accumulation is constant" (page 18, line 415).

Then, we compare the mean TOC and accumulation values of PLACE and Tunu13 records. This comparison reveals lower TOC for lower accumulation. We have rephrased as follows: "Comparing Tunu13 and PLACE TOC records suggest that the lower the accumulation rate, the lower the TOC baseline concentrations in the deep ice."

Both information, describing datasets, are relevant when investigating the link between TOC and accumulation.

L402: Clarify what you mean by "at play"? What is the driving question here? This is where I start lose track of what you are after, hence my general comment above about possibly omitting the ammonium related text and figures.

The comparison between TOC and ammonium is key as it demonstrates that lower accumulation sites can experience TOC remobilization, within the snow itself and after TOC deposition. This is a new and unexpected result, which contributes to a better understanding of the chemical processes involved in CO in situ production. This is stated as follow in Sect. 3.3.2 : "We can not rule out, however, that a redistribution of organic carbon along depth driven by OC post-deposition process (shifting the ammonium-formate equilibrium with the gas phase) impacts specifically the Tunu13 CO record by providing some additional organic substrates in winter layers"

As stated before, the sections 3.3.2 and 3.3.3 have been merged in a single section now entitled " TOC patterns and in situ CO production". This should provide more clarity to the manuscript.

The wording "at play" is not anymore used in the revised manuscript.

L423-425: The first sentence says baseline CO does not significantly correlate with mean or baseline TOC. The second sentence says there are similarities in trends and patterns of the same two quantities. The sentences are conveying conflicting messages.

Line 423-425, we stated that "the baseline CO mixing ratio at Tunu13 [...] does not exhibit significant correlations with mean or baseline TOC concentrations. However, similarities in trends and patterns can be seen when comparing Tunu13 baseline CO and TOC (Fig. 6)."

We agree that the sentences were somewhat conveying conflicting messages. Figure 6 was removed from the manuscript because the statement "similarities in trends and patterns can be seen when comparing Tunu13 baseline CO and TOC" was subjective.

About the impact of analytical smoothing on baseline levels:
L427-429: This is pretty speculative. If I could see the Tunu record being clearly biased high from the other data sets in Fig. 1, I would be fine with it.

Here the reviewer finds that the following statement is speculative : "The larger analytical smoothing impacting the Tunu13 CO record means that some of the CO baseline signal likely incorporates in situ produced CO from spring/summer ice layers."

We have compiled a code to investigate quantitatively how analytical smoothing can impact a CFA CO record. A synthetic CO record is first generated with a constant CO baseline ([CO] = 80 ppbv), and CO spikes randomly generated on top of this baseline. CO spikes are generated annually in summer layers (the length of each spike is about 2 months). This synthetic CO signal is generated on an age scale, and accumulation rate is used to report this signal on a depth scale. Finally, the synthetic CO signal reported on a depth scale is convoluted with the green function characterizing the CFA analytical smoothing (Fig. S8-11, central panels). We obtain a smoothed signal specific to an analytical campaign (site and CFA system). We extract a 5th baseline from that signal and compare it with the original constant, 80 ppbv, baseline.

This model is described in the new 1.9.2 SI section, and suggests that for the Tunu13 record, positive bias of up to 8 ppbv can impact the 5th baseline along multi-meters depth intervals. On the other hand, we do not find bias larger than ~2 ppbv for other records.

L432-433: I don't see a MAD figure in section 3.1.
A figure (Fig. S16) was added to the manuscrit, and is now referenced in this section mentioned by the reviewer.

L439-440: I'm convinced that analytical smoothing is not a factor for the PLACE baseline. For any other production process, I am not convinced PLACE baseline would not be impacted.
We applied the "smoothing code" described previously to the PLACE core (2015 SRI and 2017 IGE CFA systems), and found negligible bias of 5th baseline related to analytical smoothing.
We have evidence for the PLACE record that TOC is not remobilized after deposition, and thus low-TOC content of winter layers should not favor CO in situ production. However, we are stating multiple times in the manuscript that, while we have no evidence for CO in situ production impacting the PLACE baseline, we can not fully rule it out. Consequently, we discuss our CO paleo-reconstruction as an upper bound of past atmospheric CO burden. This is in agreement with the reviewer's comment.

L457: I do not see an extremely sharp increase in the Eurocore data.
The word "extremely" was removed from this sentence.

**Review #2 from Maria Elena Popa**

**General comments**

- It is somewhat confusing that a long records are mentioned, but the data are not shown; the paper refers sometimes to longer pieces shown only in the supplement. It would be good to have an overview, somewhere in the beginning of the results: what time period is presented in the paper for CO (and if not the whole record, explain why) and what time periods or ice depth intervals are used for various tests or secondary measurements.

We now introduce the Results and Discussion Section explaining that all records are available on the period spanning the last three centuries, and only three records extend beyond the last three centuries.

- The main results of this paper are in Sect. 3.5. This comes quite late in a complicated paper, and the reader may have lost some energy by then. Consider moving this section to the beginning of the results, and have the more technical discussion later. (it is also a bit difficult to go back from here to Fig. 1).

We understand that the manuscript was difficult to read. Section 3.3 has been largely shortened (see responses to review #1) and thus it should help the reader to reach Section 3.5.

- A more detailed discussion would be interesting on the link between the observed CO in ice and the known/ assumed history of atmospheric CO sources. A very short mention is included now in the Conclusion at line 545, but this seems to me insufficient. Are the trends observed consistent with what is known (or believed) about atmospheric CO? In particular, what caused the steady increase starting in 1970s?

We understand that the lack of atmospheric interpretation in the manuscript can be frustrating for the reader. However, we have chosen to publish such interpretation in a second paper, as stated in the introduction '"The comparison of the past evolution of Arctic atmospheric CO mixing ratio extracted from Greenland ice archives in the frame of this study with AerChemMIP model outputs is out of the scope of this paper. Such comparison, which should also allow to better constrain CO emissions inventories of emissions factors, will be addressed in a future study."

The reasons for such choice include: (i) this manuscript submitted to Climate of the Past is already very long, with a a large methodological section (SI are about 40 pages), (ii) the next study which will discuss the comparison between ice core [CO] records and AerChemMIP output will also include a CO record from Southern Hemisphere, (iii) this next study will be easier to read in the modeling community as it will not include too many analytical details.

- uncertainties: sometimes 1-sigma and sometimes 2-sigma uncertainties are reported. This is somewhat confusing, would it be possible to stick with one of them? Also, please check that they are not mixed when propagating errors.

All CO data uncertainties were reported as 1-sigma, except the internal precision which was reported as 2-sigma. Internal precision is now reported as 1-sigma, so all uncertainties reported in the revised manuscript are reported as 1-sigma.

This revision has no impact on how we propagate errors. Indeed, we use external precisions, and not internal precisions, when propagating errors. External precisions include all sources of errors or bias, notably the internal precisions and stability of gas-CFA measurements.

The DRI depth scale uncertainty was reported as 2-sigma. It is now reported as 1-sigma (Sect.2.2.8).

- If possible provide a link to the data or include the data as a supplement.

All CO dataset discussed in this study will be made available through the Pangea database (https://pangaea.de/).

**Specific comments**

Line 32: Reference "Myrhe et al. 2013": the correct spelling is "Myhre"; the reference does not appear in the reference list at the end.

The manuscript was corrected.

Line 71: typo?

It should read CH4. The manuscript was corrected.

Line 93: Table 1 shows 6 entries, please clarify if two of these refer to the same ice core

The 6 entries of Table 1 refer to 6 different ice cores. Two of them (NEEM-2011-S1, and NEEM-2011-SC) were collected at the same site, but cover different depth and age scales. For clarity, we did not modify Table 1, but we added the following sentence to the manuscript : "The NEEM-SC and NEEM-2011-S1 archives are different ice cores, but drilled at the same location."

Lines 133, 134: spell out DRI and IGE at the first use

The manuscript was corrected.

Line 176: I did not understand how exactly the external precision was calculated: what is "pooled standard deviation calculated on the differences", is this the standard deviation of the differences? Or the standard deviation of the normal distribution that would result in the measured differences, when sampling randomly two data points? Or something else? Please clarify.

After binning the main (M) data and replicate (R) data into few cm long intervals, we obtain $n$ duplicate measurements. The pooled standard deviation is then calculated as follow :

$$\sqrt{\left(\left(\frac{\sum(M-R)^2}{2 \times n}\right)\right)}$$

This is now written in the SI (Sect. 1.7).

Line 187: How is the link to the WMO-CO X2014A calibration scale established? Is it via the same cylinders described in 2.3? If yes, better to mention them already here, and also send to the more detailed description in SI.
We have modified the manuscript (Sect. 2.2.6) and the SI (Sect. 1.8.2) to specify that the cylinders used for CFA calibration were the same as those used for SARA calibrations. These cylinders are described in Table S3.

Line 253: I suggest removing the drilling information from here and including it all in Table 1
Table 1 presents the drillings investigated for continuous CO measurements.
Line 255 (page 10), we describe specific samples analyzed with a different analytical methodology to investigate rapid CO production from trace organics in the ice during melting.
In our opinion, it would be confusing to describe all these samples in the same table. We already have modified Table 1 to add drilling years as suggested by the reviewer, which already makes table 1 a bit larger.

Line 288: "Median Average Deviation" - should this be "Median Absolute Deviation"?
This is correct, the manuscript was corrected.

Lines 288-296: MAD is discussed here, but it is not shown anywhere - consider showing it in the supplement
A new figure showing MAD was added to SI (Fig. S18).

Line 291: are these really integrated values? then the unit should include a time component. Otherwise remove the word "integrated"
The manuscript was modified as follow : "MAD values averaged over the entire 1700-1950 CE period range 9.9-14.2 ppbv, depending on the records"

Line 320 and Fig. S12: it actually looks like the minima in the initial data are systematically lower than in the replicate, by 5-10 ppb. Considering that the baseline is the main subject of this paper, it seems relevant.
*Figure S12 is now Fig. S14 in the revised manuscript.*
We agree that Fig. S14 suggests lower minima for the initial data (i.e., collected in Feb. 2017). Looking closely at Fig. S14, we have identified 24 minimum CO values. The initial data was

always lower on these minimums. However, for 21 events, the difference between initial and replicate CO mixing ratios is lower than 8 ppbv. The combined uncertainty for each dataset is 4.1 ppbv (1-sigma). Thus we can not conclude here that initial [CO] are significantly lower, but instead datasets overlap within their uncertainty enveloppes.

We observed two events (resp. one event) with a [CO] difference between initial and replicate measurements of 9 ppbv (resp. 10 ppbv). The average (+/- 1 standard deviation) difference between initial and replicate measurements calculated over the 24 events is 6.1 +/-2.2 ppbv.

Line 368 and Figs. 2, 4: since TOC and NH4 are only discussed in 3.3.1, is it possible to let Fig. 2 focused on the resolution comparison between DRI and IGE, and move the PLACE TOC and NH4 data to Fig. 4?

The TOC and NH4 data from PLACE core, shown in Fig. 2 are, indeed, only discussed in Sect. 3.3. However, we prefer not reorganizing Fig. 2 and 4, because we do compare CO with TOC, and NH4 Place records in the manuscript.

Lines 455-458: the description of the differences between Eurocore discrete data (blue dots) and PLACE baseline does not seem consistent with the figure, please check the years. (e.g. the two datasets seem to diverge around 1830, there is no Eurocore datapoint around 1905, the increase rate in Eurocore baseline is approx. constant starting from ~1870, etc)

A plotting error concerning the x-scale (age-scale) of Fig. 7 was corrected. This error (a 6 yrs constant offset applying to all data shown, due to a wrong transfer from depth to age) does not change our discussions and interpretation. The Eurocore data is now reported in 1905, as described by Haan et al. (1996).

Lines 518-519: more details may be needed here. What is ACCMIP? Does the "[CO] output" refer to atmospheric CO simulated by a model? What time scales are included here?

We revised the manuscript to explain that ACCMIP is the Atmospheric Chemistry and Climate Model Intercomparison Project. We have rephrased this paragraph to remove "[CO] output" which is a vocabulary specific to modelling studies. We now highlight better that the time scale included here are the years 1980 and 2010.

Fig. 7: the green and blue point markers are difficult to distinguish in the figure - consider changing the color of the green markers

*Fig. 7 is now Fig. 6 in the revised manuscript.*

Color and shape of markers were changed on Fig. 6.

Table 1: I suggest including the drilling year

Drilling years were added to Table 1.

Sect. 3.3 is now entitled "Atmospheric CO history retrieved from Greenland ice cores".

**References :**

Faïn, X., Chappellaz, J., Rhodes, R. H., Stowasser, C., Blunier, T., McConnell, J. R., Brook, E. J., Preunkert, S., Legrand, M., Debois, T. and Romanini, D.: High resolution measurements of carbon monoxide along a late Holocene Greenland ice core: evidence for in situ production, Clim. Past, 10(3), 987–1000, doi:10.5194/cp-10-987-2014, 2014.

Haan, D., Martinerie, P. and Raynaud, D.: Ice core data of atmospheric carbon monoxide over Antarctica and Greenland during the last 200 years, Geophys. Res. Lett., 23(17), 2235–2238, 1996.

Novelli, P. C., Masarie, K. A. and Lang, P. M.: Distributions and recent changes of carbon monoxide in the lower troposphere, J. Geophys. Res., 103(D15), 19015–19033, 1998.

Petrenko, V. V., Martinerie, P., Novelli, P. C., Etheridge, D. M., Levin, I., Wang, Z., Blunier, T., Chappellaz, J., Kaiser, J., Lang, P., Steele, L. P., Hammer, S., Mak, J., Langenfelds, R. L., Schwander, J., Severinghaus, J. P., Witrant, E., Petron, G., Battle, M. O., Forster, G., Sturges, W. T., Lamarque, J.-F., Steffen, K. and White, J. W. C.: A 60 yr record of atmospheric carbon monoxide reconstructed from Greenland firn air, Atmos. Chem. Phys., 13(15), 7567–7585, doi:10.5194/acp-13-7567-2013, 2013.

---

## Author Response (AR2)

**Response to Referees – cp-2019-94**

We are extremely thankful to Murat Aydin for the time he dedicated to a second round of review, and his constructive comments on the article. We listed below our responses to the major and minor specific comments. The comments of the referee are in black, and our corresponding responses are below in blue. All the section and figure numbering included in our response to reviewers refers to the updated manuscript and SI.

Xavier Faïn on behalf of all co-authors

**REVIEW FROM MURAT AYDIN**

One of my general comments was about the possibility of firn processes having something to do with the production of CO spikes. I gather from their responses that the authors disagree and see this as a process happening within the ice matrix after bubble closure. It may be a good idea to include a sentence to this effect to orient the reader.

Petrenko et al. (2013) have investigated the occurrence of in situ CO production in the firn air at three Greenland sites. They cannot fully exclude such in situ production, and we already report in the manuscript that "a small contribution from in situ CO production (up to 5 ppbv) within the firn itself could not be ruled out (Petrenko et al., 2013)" (Sect 3.5.2). In our study, we did not identified any reasons that contradict this conclusion from Petrenko et al. (2013). This does mean that in situ production is a process happening in the ice matrix after bubble closure. To clarify the manuscript, we have modified Sect 3.5.2 and the conclusion to mention this.

I understand the statement about not having any direct evidence for CO production happening in the firn. However, some of the observations reported here are very difficult to explain solely with production after bubble closure. Specifically, there is no straight forward interpretation of the CO vs. the corresponding TOC data. New figures show seasonality in TOC and CO in the PLACE ice core, but the TOC in summer and winter layers appears to be within 10-20% of the mean levels and the lack of overall correlation between TOC and CO is puzzling.

The "summer" and "winter" TOC data reported on Fig. S26 are indeed within 20% of the mean levels, but we have defined summer (resp. winter) as April-September (resp. October-March). The TOC values extracted over these periods lasting 6 months are lower (resp. higher) than summer maximum (resp. winter minimum). However, extracting data at higher resolution was not possible considering the accuracy of the depth scale.

Note: the Fig. S26 has been slightly modified as we found an error in the plot.

Thinking purely from this perspective, I don't see why there would be no in situ production in the winter layers, for example.

We agree that in situ production in winter layers cannot be fully excluded, this is why we recommend to interpret our multisite CO reconstruction as an upper bound of past atmospheric CO abundance in the Arctic atmosphere.

Finally, we have additional reasons to consider that in situ CO production can occur in winter layer at the Tunu13 site (because of remobilization of TOC from summer to winter layers). Such reasons partially explain why we have not considered Tunu13 in the multisite composite.

I agree with the need to better understand the fate of TOC in the firn and snow, but does this not imply that firn processes could have a role in the production of CO?
Yes, we identify one specific firn process that can have a role in the production of CO at sites of low accumulation: TOC remobilization from summer layers to winter layers (Sect. 3.3.2). For this reason, we have conservatively excluded Tunu13 from the multisite composite.

The discussions regarding the fate of TOC and the relationship with $NH_4^+$ are snow/firn processes that the readers will struggle to connect to CO production in the ice after bubble closure. Perhaps, there is more that can be said to help make this connection?
In the summary section 3.3.3, we do say that "we cannot rule out that a redistribution of organic carbon along depth driven by OC post-deposition process (shifting the ammonium-formate equilibrium with the gas phase) impacts specifically the Tunu13 CO record by providing some additional organic substrates in winter layer". With this sentence, we explain why the fate of TOC, revealed by the TOC-$NH_4^+$ relationship, can impact CO production. We have modified Section 3.3.2 to better explain this point.

In my opinion, the strongest argument for accepting the composite 5th percentile of measurements as a likely atmospheric record is the agreement between different ice core measurements and firn records. The paper should emphasize this angle in the abstract and in the conclusions.
We have modified the abstract and conclusion accordingly.

The authors chose not to include the Tunu record in the final composite history. My whole point in arguing for its inclusion was that this would not significantly alter the final composite record. The conceptual model included in the supplement (Fig. S12) is pretty helpful in demonstrating how the analytical smoothing impacts the measurements, but the average increase in CO is always less than 10 ppb and 3-4 ppb on average. The figures they included in the review also show that the composite record looks pretty much the same with or without inclusion of the Tunu record. In the end, I don't really mind leaving out the Tunu record. If they believe there is potential for recovering a long term record at Tunu, it can be pointed out somewhere that artifacts due to analytical smoothing do not seem to drive the type of large long term trends apparent in the composite record and show Fig. R4 in the supplement.
We did not fully understand the point that the reviewer made in the last sentence of the previous paragraph, and thus we were not able to address it (what is Fig. R4?). Overall, the revised manuscript does not include any comments on the potential of the Tunu13 site for recovering a long term record. Such recovery would require improved analytical capability (i.e., a better resolution), and seems speculative at this point. However, the full Tunu13 CO record is provided (Fig. 17), and future studies will be able to start new investigation using this dataset.

**Line by line comments/suggestions:**

L13-15: This sentence gives the impression that the co-examination of CO and TOC records shows there is no CO production in winter layers. The strongest evidence for winter layers containing largely an

atmospheric signal, hence the path towards recovering an atmospheric record from 5th percentile baselines, comes from the agreement between different ice core records in my opinion. I cannot readily infer from this that TOC in the winter layers does not produce CO in the ice while it does in summer layers because no possible explanation is provided for why it should not.

We agree with the reviewer and have modified the abstract. We kept the statement that, in our study, higher accumulation easier the extraction of atmospheric history from continuous CO signal. We added that the agreement in CO baseline between different Greenland sites supports that winter layers do contain an atmospheric relevant information. The good overlap between the multisite ice core reconstruction and the firn air CO history was already reported in the abstract.

Note: we have removed the wording "conducted at Summit" in sect. 3.4, as the Greenland firn air reconstruction is based on 3 sites (as discussed in sect 3.5.2).

L85: "Burden" is commonly used to mean the total mass/moles of a gas in the atmosphere. Better to use mixing ratio, abundance, or levels instead.

"Burden" has been replaced by "levels" or "abundance" in different sections of the manuscript, including line 85.

L129: Rephrase "extracted along the sample line…"

We rephrased as follow: "the gas is recovered from the sample line…"

L291: Ice core trends are not strictly "monotonic" even after 1875, especially for some of the cores; might be better to say "steady."

We are now using "steady" instead of "monotonic".

L297: S17 instead of S15.

The manuscript has been corrected.

L324: Consider replacing "never expose directly freshly drilled cores to sunlight" with "never expose freshly drilled cores to direct sunlight."

The manuscript has been corrected accordingly.

Lines 345-365, section 3.2.4: This 6 ppb appears to be a system blank. Is it subtracted from the discrete measurements? I could not see this info anywhere. It can probably be added to the caption of Fig. 3.

These 6 ppbv could indeed be considered as a system blank for the discrete CO analyses discussed in Sect. 3.2.4. However, in this section we investigate if in extractu CO production could occur during the melting process. Thus, the data were not corrected for blank as our goal was to observe is such blank would be increasing with time.

On the other hand, the CO data collected with the discrete method and used for a comparison with CFA dataset had to be corrected for the system blank. This is now clearly stated in the SI (Sect. SI 1.8.4), as follow : "Blank corrections were applied by subtracting the average of four gas free ice runs (CO mole fraction = 9.0 ± 2.1 ppbv) that were run concurrently with the five PLACE core samples. The CO mole

fraction measurement uncertainty for the five PLACE cores samples was defined as the CO mole fraction variability of the gas free ice measurements."

Line 426-427: The hypothesis here is not easy to understand. Do you mean only a fraction of TOC converts to CO and the excess above a certain threshold does not result in more CO production?
Lines 426-427 read: "The larger analytical smoothing impacting the Tunu13 CO record means that some of the CO baseline signal likely incorporates in situ produced CO from spring/summer ice layers". We are not sure how this statement relate to the comment from the reviewer.

Line 450: Incorporating instead of "incorporates."
The manuscript has been corrected accordingly.

Line 453: No need for "however."
The manuscript has been corrected accordingly.

Line 467-468: I think you should put the baseline back in the figure otherwise it is hard to see this. You can show a version with a mean in the supplement.
The CFA PLACE baseline is difficult to be included in Fig. 6. On one side this would allow a comparison with the historical Eurocore data from Haan et al. (1998). On the other hand tjis would be confusing as such baseline can not be compared directly to discrete PLACE record (which should be compared to the mean CFA signal). This issue was discussed in the first round of review.
To address the comment raised by the reviewer, we have added the specific figure shown below in the SI which plots only CFA PLACE and historical Eurocore CO dataset. This new figure is now referred in Sect.

[Figure]

Figure S23. Continuous CO mixing ratio collected along the Place ice core (grey line) with 5th percentile baseline (black line and envelop), and historical Eurocore discrete CO data (blue dot, Haan et al 1998).

Section 3.5.1: There is too much emphasis on Fig. 1, which has already been discussed. It would be better to point out the trends in Fig. 7, which is done in the conclusions but seems out of place there.
We are now emphasizing better Fig. 7 in section 3.5.1 with the sentence "Overall, Fig. 7 reports a ~30% increase in CO concentration at high latitudes of the northern hemisphere", which was previously referring to Fig. 1. We kept the first paragraph of Sect 3.5.1 discussing Fig. 1 as it is important to discuss the agreement between different ice core measurements and firn records (see general comments from the reviewer).